# Seasonal Variability of Stratospheric Methane: Implications for Constraining Tropospheric Methane Budgets Using Total Column Observations

K. M. Saad[1], D. Wunch[1,2], N. M. Deutscher[3,4], D. W. T. Griffith[3], F. Hase[5], M. De Mazière[6], J. Notholt[4], D. F. Pollard[7], C. M. Roehl[1], M. Schneider[5], R. Sussmann[8], T. Warneke[4], and P. O. Wennberg[1]

[1]California Institute of Technology, Pasadena, California, USA
[2]University of Toronto, Toronto, Ontario, Canada
[3]University of Wollongong, Wollongong, NSW, Australia
[4]University of Bremen, Bremen, Germany
[5]Karlsruhe Institute of Technology, IMK-ASF, Karlsruhe, Germany
[6]Royal Belgian Institute for Space Aeronomy, Brussels, Belgium
[7]National Institute of Water and Atmospheric Research, Omakau, New Zealand
[8]Karlsruhe Institute of Technology, IMK-IFU, Garmisch-Partenkirchen, Germany

*Correspondence to:* K. M. Saad (katsaad@caltech.edu)

**Abstract.** Global and regional methane budgets are markedly uncertain. Conventionally, estimates of methane sources are derived by bridging emissions inventories with atmospheric observations employing chemical transport models. The accuracy of this approach requires correctly simulating advection and chemical loss such that modeled methane concentrations scale with surface fluxes. When total column measurements are assimilated into this framework, modeled stratospheric methane introduces additional potential for error. To evaluate the impact of such errors, we compare Total Carbon Column Observing Network (TCCON) and GEOS-Chem total and tropospheric column-averaged dry-air mole fractions of methane. We find that the model's stratospheric contribution to the total column is insensitive to perturbations to the seasonality or distribution of *tropospheric* emissions or loss. In the Northern Hemisphere, we identify disagreement between the measured and modeled stratospheric contribution, which increases as the tropopause altitude decreases, and a temporal phase lag in the model's tropospheric seasonality driven by transport errors. Within the context of GEOS-Chem, we find that the errors in tropospheric advection partially compensate for the stratospheric methane errors, masking inconsistencies between the modeled and measured tropospheric methane. These seasonally-varying errors alias into source attributions resulting from model inversions. In particular, we suggest that the tropospheric phase lag error leads to large misdiagnoses of wetland emissions in the high latitudes of the Northern Hemisphere.

## 1 Introduction

Identifying the processes that have driven changes in atmospheric methane ($CH_4$), a potent radiative forcing agent and major driver of tropospheric oxidant budgets, is critical for understanding future impacts on the climate system. Methane's growth rate, which had been decreasing through the 1990s from about 10 to 0 ppb per year, began to increase again in 2006 and over

the past decade has averaged 5 ppb per year (Dlugokencky et al., 2011). Developing robust constraints on the global $CH_4$ budget is integral for understanding which processes produced these decadal trends (e.g., Bergamaschi et al., 2013; Wecht et al., 2014a, b; Turner et al., 2015).

One common approach to quantifying changes in the spatial distribution of sources are atmospheric inversions, which incorporate surface fluxes estimated by bottom-up inventories as boundary conditions for a chemical transport model (CTM). The modeled $CH_4$ concentrations are compared to observations within associated grid boxes, and prior emissions are scaled to minimize differences with measured dry-air mole fractions (DMFs), producing posterior estimates. The accuracy of these optimized emissions depends on how well the CTM simulates atmospheric transport and $CH_4$ sinks, which are generally prescribed.

Pressure-weighted total column-averaged DMFs ($X_{gas}$) provide a relatively new constraint and have previously been shown to improve estimates of regional and interhemispheric gradients in trace gases (Yang et al., 2007). Fourier transform infrared spectrometers can measure $CH_4$ DMFs ($X_{CH_4}$) from ground-based sites, such as those in the Total Carbon Column Observing Network (TCCON) and Network for the Detection of Atmospheric Composition Change (NDACC), and satellites, including SCanning Imaging Absorption spectroMeter for Atmospheric CartograpHY (SCIAMACHY) (Bergamaschi et al., 2007), Greenhouse gases Observing SATellite (GOSAT) (Parker et al., 2011), and the upcoming TROPOspheric Monitoring Instrument (TROPOMI) (Butz et al., 2012). These observations complement surface measurements because they add information about the vertically-averaged profile and are sensitive in the free troposphere (Yang et al., 2007). Additionally, they complement aircraft observations by measuring trace gases at higher temporal frequency, although they share the limitation of not measuring in inclement weather. Satellite measurements add global coverage that can fill in gaps where in situ observations are sparse. Fraser et al. (2013) found assimilating GOSAT $CH_4$ columns into the GEOS-Chem CTM with an ensemble Kalman filter reduced posterior emissions uncertainties by $9-48\%$ for individual source categories and by more than three times those of inversions that only assimilated surface data for most regions. Wecht et al. (2014b) determined from their analysis of observing system simulation experiments (OSSEs) that TROPOMI's daily frequency and global coverage performs similarly to aircraft campaigns on sub-regional scales and could provide a constraint on California's $CH_4$ emissions similar to CalNex aircraft observations (Santoni et al., 2014; Gentner et al., 2014).

Incorporating total columns into modeling assessments can also be used to diagnose systematic issues with model transport. For example, comparing carbon dioxide ($CO_2$) from TCCON and TransCom (Baker et al., 2006), Yang et al. (2007) found that most models included in the comparison lack sufficiently strong vertical exchange between the planetary boundary layer (PBL) and the free troposphere, thereby dampening the seasonal cycle amplitude of $X_{CO_2}$. The limitations of models to accurately represent vertical transport can lead to radically different spatial distributions of fluxes; Stephens et al. (2007) found, for example, that the northern terrestrial carbon land sink and tropical emissions were overestimated by 0.9 and 1.7 PgC·year$^{-1}$, respectively, when constraining models with aircraft $CO_2$ profiles. More recent studies attribute to model transport errors the tendency of simulated $CH_4$ in the Southern Hemisphere to be higher at the surface than the free troposphere, in contrast with measurements (Fraser et al., 2011; Patra et al., 2011).

Tropospheric $CH_4$ typically does not vary radically with height above the PBL; above the tropopause, however, the vertical profile of $CH_4$ exhibits a rapid decline with altitude as a result of its oxidation and the lack of any source beyond advection from the troposphere. Fluctuations in stratospheric dynamics, including the height of the tropopause, change the contribution of the stratosphere to the total column. $CH_4$ profiles with similar tropospheric values can thus have significant differences in $X_{CH_4}$ (Saad et al., 2014; Washenfelder et al., 2003; Wang et al., 2014).

Provided that simulations replicate seasonal and zonal variability of stratospheric $CH_4$ loss, tropopause heights, and vertical exchange across the upper troposphere and lower stratosphere (UTLS), posterior flux estimates from inversions incorporating $X_{CH_4}$ measurements would not be sensitive to stratospheric processes. However, most models do not accurately represent stratospheric transport, producing low age of air values and zonal gradients in the subtropical lower stratosphere that are less steep than observations (Waugh and Hall, 2002). The TransCom-$CH_4$ CTM intercomparison assessment of transport using sulfur hexafluoride $SF_6$ showed a strong correlation between the stratosphere-troposphere exchange (STE) rate and the model's $CH_4$ budget and a weaker correlation between the $CH_4$ growth rate and vertical gradient in the model's equatorial lower stratosphere (Patra et al., 2011). These forward model dependencies of $CH_4$ concentrations to vertical transport, both within the troposphere and across the tropopause, have the potential to introduce substantial errors in atmospheric inversions. As temporal and spatial biases in a model's vertical profile will alias into posterior emissions, inversions that incorporate total column measurements must ensure that the stratosphere is sufficiently well described so as to not introduce spurious seasonal, zonal and interhemispheric trends in $CH_4$ concentrations and consequently emissions.

In this analysis, we identify systematic model errors in the seasonal cycle and spatial distribution of $CH_4$ DMFs by comparing TCCON total and tropospheric columns (Saad et al., 2014) to vertically integrated profiles derived from the GEOS-Chem CTM (Bey et al., 2001; Wang et al., 2004; Wecht et al., 2014a). We assess the impact of errors in the characterization of stratospheric processes on the assimilation of $X_{CH_4}$ and resulting posterior emissions estimates. In Section 2 we describe the TCCON column measurements and GEOS-Chem set up and characteristics. In Section 3 we present the results of the measurement-model comparison. In Section 4 we compare the base case simulation to one in which emissions fluxes do not vary within each year and quantify the sensitivity of source attribution of the biggest seasonal emissions sector, wetlands, to the tropospheric seasonal delay.

## 2 Methods

### 2.1 Tropospheric Methane Columns

TCCON has provided precise measurements of $X_{CH_4}$ and other atmospheric trace gases for over ten years (Wunch et al., 2011a, 2015). Developed to address open questions in carbon cycle science, the earliest sites are located in Park Falls, Wisconsin, United States and Lauder, New Zealand at about $45°$ North and South, respectively. Since 2004, the ground-based network of Fourier transform spectrometers has expanded greatly. $X_{CH_4}$ are processed with the current version of the TCCON software, GGG2014, to be consistent, and thereby comparable, across sites. Total column retrievals are generated with the GFIT nonlinear least-squares fitting algorithm, which calculates the best spectral fit of the solar absorption signal to an a

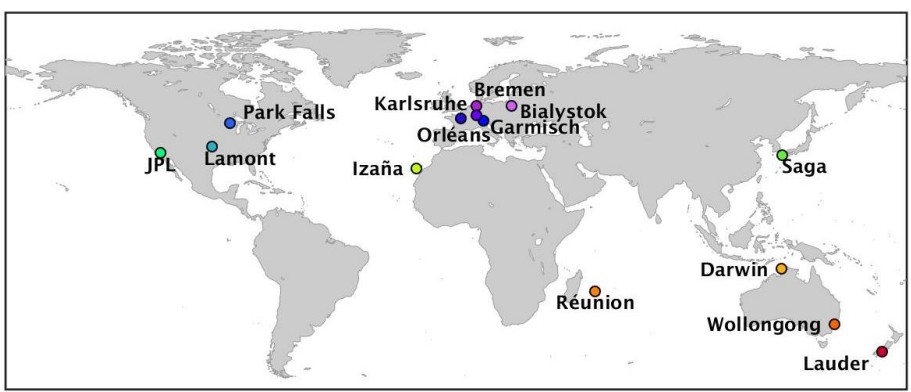

**Figure 1.** Map of TCCON sites used in this analysis. Site colors are on a spectral color scale in order of latitude, with Northern Hemisphere sites designated by cool colors and Southern Hemisphere sites designated by warm colors.

priori vertical profile and outputs a vertical scaling factor. The pressure-weighted integration of the scaled a priori profile produces column abundances, which are then divided by the dry air column, calculated using concurrently retrieved oxygen ($O_2$) columns (Wunch et al., 2010, 2011a, 2015). Trace gas a priori profiles are derived with empirical models, which are generated incorporating aircraft and balloon in situ and satellite measurements (see Wunch et al., 2015, for a complete list), and for $CH_4$

include a secular increase of 0.3% per year and an interhemispheric gradient in the altitude dependence of the vertical profiles (Toon and Wunch, 2014). These models are fit to daily noontime National Centers for Environmental Protection and National Center for Atmospheric Research (NCEP/NCAR) reanalysis pressure grids (Kalnay et al., 1996), interpolated to the surface pressure measured real-time on site. Because the profile of $CH_4$ drops off rapidly in the stratosphere, the accuracy of the a priori shape, and thus the retrieved column, depends on correctly determining the tropopause.

Tropospheric columns have been shown to represent the magnitude and seasonality of in situ measurements (Saad et al., 2014; Washenfelder et al., 2003; Wang et al., 2014). The tropospheric $CH_4$ column-averaged DMFs ($X_{CH_4}^t$) are derived by the HF-proxy method described in Saad et al. (2014), which uses the relationship between $CH_4$ and HF in the stratosphere, derived from ACE-FTS satellite measurements (Bernath, 2005; De Mazière et al., 2008; Mahieu et al., 2008; Waymark et al., 2014), to calculate and remove the stratospheric contribution to $X_{CH_4}$. The $X_{CH_4}^t$ used in this analysis have been processed consistently

with the GGG2014 TCCON products, with airmass dependence and calibration factors calculated for and applied to $X_{CH_4}^t$ (Wunch et al., 2010, 2015). Additional details about the tropospheric $CH_4$ measurements can be found in Appendix A.

    With the exception of Eureka and Sodankylä, which are highly influenced by the stratospheric polar vortex, all TCCON sites that provide measurements before December 2011 are included in this analysis (Fig. 1). Table 1 lists locations and data collection start dates for each of the sites.

**Table 1.** TCCON sites, coordinates, altitudes, start date of measurements and locations used in this analysis.

| Site | Latitude (°) | Longitude (°) | Elevation (km) | Start Date | Location | Data Reference |
|------|------|------|------|------|------|------|
| Bialystok | 53.2 | 23.0 | 0.18 | Mar 2009 | Bialystok, Poland | Deutscher et al. (2014) |
| Bremen | 53.1 | 8.9 | 0.03 | Jan 2007 | Bremen, Germany | Notholt et al. (2014) |
| Karlsruhe | 49.1 | 8.4 | 0.11 | Apr 2010 | Karlsruhe, Germany | Hase et al. (2014) |
| Orleans | 48.0 | 2.1 | 0.13 | Aug 2009 | Orleans, France | Warneke et al. (2014) |
| Garmisch | 47.5 | 11.1 | 0.75 | Jul 2007 | Garmisch, Germany | Sussmann and Rettinger (2014) |
| Park Falls | 45.9 | -90.3 | 0.47 | Jan 2005 | Park Falls, WI, USA | Wennberg et al. (2014b) |
| Lamont | 36.6 | -97.5 | 0.32 | Jul 2008 | Lamont, OK, USA | Wennberg et al. (2014c) |
| JPL | 34.2 | -118.2 | 0.39 | Jul 2007 | Pasadena, CA, USA | Wennberg et al. (2014d, a) |
| Saga | 33.2 | 130.3 | 0.01 | Jul 2011 | Saga, Japan | Kawakami et al. (2014) |
| Izaña | 28.3 | -16.5 | 2.37 | May 2007 | Tenerife, Canary Islands | Blumenstock et al. (2014) |
| Darwin | -12.4 | 130.9 | 0.03 | Aug 2005 | Darwin, Australia | Griffith et al. (2014a) |
| Réunion Island | -20.9 | 55.5 | 0.09 | Sep 2011 | Saint-Denis, Réunion | De Maziere et al. (2014) |
| Wollongong | -34.4 | 150.9 | 0.03 | Jun 2008 | Wollongong, Australia | Griffith et al. (2014b) |
| Lauder | -45.0 | 169.7 | 0.37 | Jan 2005 | Lauder, New Zealand | Sherlock et al. (2014a, b) |

## 2.2 GEOS-Chem Model

Model comparisons use the offline $CH_4$ GEOS-Chem version 9.02 at $4° \times 5°$ horizontal resolution on a reduced vertical grid (47L). $CH_4$ loss is calculated on 60 minute intervals and is set by annually-invariable monthly 3D fields: hydroxyl radical (OH) concentrations in the troposphere (Park et al., 2004) and parameterized $CH_4$ loss rates per unit volume in the strato-
5 sphere (Considine et al., 2008; Allen et al., 2010; Murray et al., 2012). Emissions are released at 60-minute time steps and are provided by the GEOS-Chem development team for 10 sectors: gas and oil, coal, livestock, waste, biofuel, and other anthropogenic annual emissions from EDGAR v4.2 (European Commission Joint Research Centre, Netherlands Environmental Assessment Agency, 2011; Wecht et al., 2014a); other natural annual emissions from (Fung et al., 1991); rice agriculture (European Commission Joint Research Centre, Netherlands Environmental Assessment Agency, 2011) and wetland (Pickett-
10 Heaps et al., 2011) monthly emissions, which incorporate GEOS5 annual and monthly mean soil moisture values; and biomass burning daily emission from GFED3 estimates (Mu et al., 2011; van der Werf et al., 2010). Loss via soil absorption (Fung et al., 1991), set annually, is subtracted from the total emissions at each time step.

### 2.2.1 Model Set Up

We initialized zonal $CH_4$ distributions with GGG2014 data version a priori profiles (Wunch et al., 2015) produced at hori-
15 zontal grid centers, which we adjusted vertically to match the zonally averaged daily mean model's tropopause, derived from the National Aeronautics and Space Administration Global Modeling and Assimilation Office (NASA/GMAO) Goddard Earth

Observing System Model, Version 5 (GEOS5). The model was run from December 2003, the first month in which GEOS5 meteorological data were available, to June 2004, the beginning of the TCCON time series; we then ran the model repeatedly over the June 2004-May 2005 time frame, which allowed us to make comparisons with the TCCON data at Park Falls and Lauder, until $CH_4$ concentrations reached equilibrium. A number of perturbation experiments were run in this way to quan-

5 tify the sensitivity of $CH_4$ distribution and seasonality to the offline OH fields, prescribed emissions, and tropopause levels (Table 2). These model experiments are described in greater detail in Appendix B1.

Using $CH_4$ fields for 1 January 2005 from the equilibrium simulation as initial conditions, model daily mean $CH_4$ mole fractions were computed through 2011. These were converted to dry mole fractions, as described in Appendix B2. In addition to the default emissions scheme, an aseasonal simulation setup, in which rice, wetland, and biomass burning emissions were

10 disabled and aseasonal emissions scaled up such that total annual zonal fluxes approximate those in the base simulation, was similarly run to equilibrium and used as initial conditions for the 2005-2011 run.

For comparisons with column measurements, model vertical profiles were smoothed with corresponding TCCON $CH_4$ averaging kernels, interpolated for the daily mean solar zenith angles, and prior profiles, scaled with daily median vertical scaling factors and interpolated to the daily mean surface pressures measured at each site, following the methodology in

Rodgers and Connor (2003) and Wunch et al. (2010). Tropospheric columns were integrated in the same manner as the total columns up to the grid level completely below the daily mean tropopause, consistent with how GEOS-Chem partitions the atmosphere in the offline $CH_4$ simulation. To test the dependence of our results on the chosen vertical integration level, tropospheric columns were also calculated assuming the tropopause was one and two grid cells above this level. While $X_{CH_4}^t$ changed slightly, by a median of about 1 and 5 ppb for a one and two-level increase respectively, shifting the tropopause did

not alter the findings discussed in this paper. A description of the model smoothing methodology and assumptions is provided in Appendix B3. The stratospheric contribution to the total column, which is calculated as the residual between the $X_{CH_4}^t$ and $X_{CH_4}$, is the amount by which the stratosphere attenuates $X_{CH_4}$ via stratospheric loss and transport (see Appendix C for the derivation).

### 2.2.2 Model Features

The seasonal amplitude of the differences between base and aseasonal simulations are small–within $\pm 4$ ppb–for all vertical levels in the Southern Hemisphere (Fig. 2). In the Northern Hemisphere, however, the difference is much larger and primarily impacts the troposphere, where it varies between $-10$ and $+13$ ppb. The insensitivity of the stratosphere to the seasonality

**Table 2.** Sensitivity Experiments

| Run Name | Description | $CH_4$ Lifetime (years) | Final $CH_4$ Burden (Tg) |
| --- | --- | --- | --- |
| Base | Default OH and Emissions | 9.55 | 4825 |
| Aseasonal | Constant Monthly Emission Rates | 9.57 | 4872 |
| Updated OH | Monthly OH fields from Standard Chemistry + Biogenic VOCs | 8.53 | 4828 |

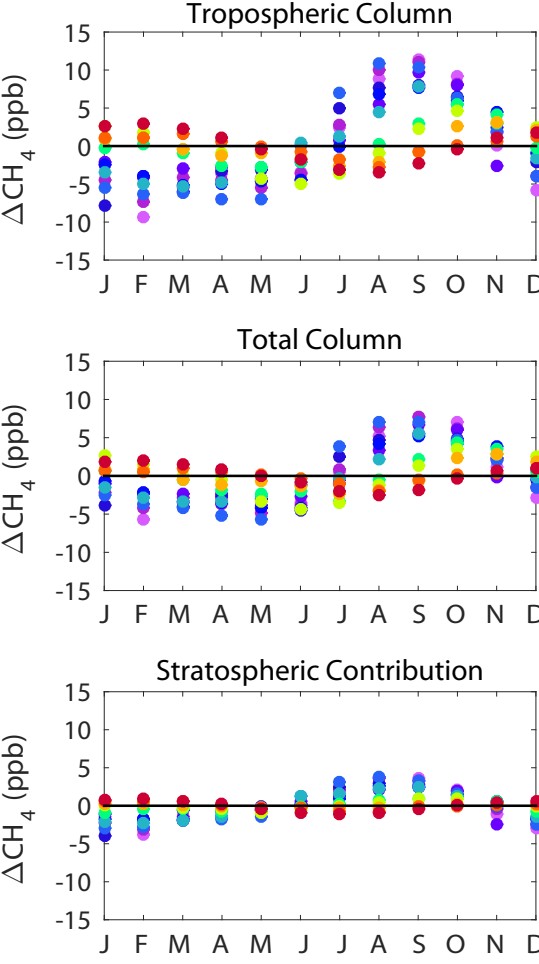

**Figure 2.** Seasonality of the difference between base and aseasonal $CH_4$ for tropospheric, total and stratospheric contribution to total columns. Site colors are as in Fig. 1.

of emissions is due to the common source of stratospheric air in the tropics (Boering et al., 1995) and the loss of seasonal information as the age of air increases (Mote et al., 1996).

Due to the relatively short photochemical lifetime of $CH_4$ in the stratosphere, about 22 months in the base simulation, stratospheric $CH_4$ concentrations stabilize much more quickly than in the troposphere (Fig. 3a). This rapid response time of the stratosphere occurs regardless of perturbations to the troposphere, such as the seasonality of emissions (Fig. 3b) or tropospheric OH fields (Fig. 3c). In both hemispheres the differences between the base and experimental simulations asymptotically approach steady state with seasonal variability over a decade in the troposphere, but oscillate seasonally around a constant mean in the stratosphere. Stratospheric differences between simulations are considerably smaller than the seasonal amplitude of the base run: within six and one ppb, respectively, versus a seasonal range of 30 ppb at Park Falls. By contrast, $X_{CH_4}^t$ have

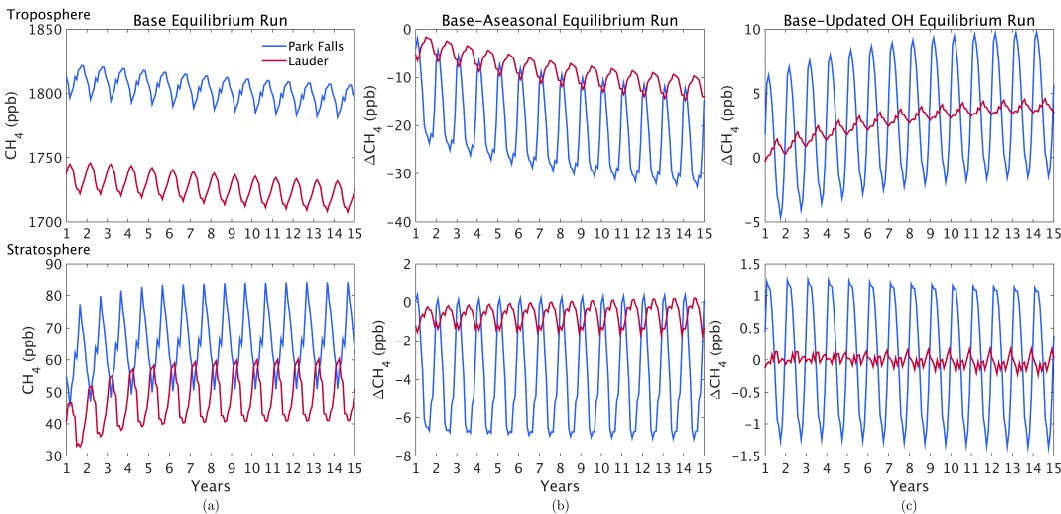

**Figure 3.** Smoothed daily mean $X_{CH_4}^t$ and stratospheric contribution to $X_{CH_4}$ at Park Falls (blue) and Lauder (red) for (a) base equilibrium simulation and the difference between the base and (b) aseasonal and (c) updated OH simulations.

differences within 30 and 10 ppb, respectively, versus a seasonal range of 20 ppb at Park Falls. The stratosphere at Lauder is even less sensitive to tropospheric perturbations.

## 3  Measurement-Model Comparison

The TCCON daily median and GEOS-Chem daily mean $CH_4$ column-averaged DMFs demonstrate a strong interhemispheric
difference for $X_{CH_4}^t$ and $X_{CH_4}$ in the both the base and aseasonal simulations (Fig. 4). The Northern Hemisphere $X_{CH_4}^t$ slope deviates from the one-to-one line more than the $X_{CH_4}$ slope ($0.60 \pm 0.02$ versus $0.86 \pm 0.03$), and the correlation coefficients are equivalent ($R^2 = 0.41$), which indicates that the poorer agreement between measurements and models in the troposphere drive the scatter in the total column.

The stratospheric contribution comparison between TCCON and the base simulation for the Northern Hemisphere sites has
an equivalent slope ($0.60 \pm 0.1$) and higher correlation coefficient ($R^2 = 0.68$) compared to $X_{CH_4}^t$ (Fig. 4c). GEOS-Chem's larger stratospheric contribution to the total column, coupled with lower tropospheric values, depresses $X_{CH_4}$. Because this effect on $X_{CH_4}$ occurs more at higher latitudes, zonal errors in the model's stratosphere balances those in the troposphere. The result is better measurement-model agreement in the total columns.

The aseasonal simulation produces lower slopes and correlation coefficients for, $X_{CH_4}^t$ (slope=$0.42 \pm 0.02$, $R^2 = 0.32$),
$X_{CH_4}$ (slope=$0.60 \pm 0.03$, $R^2 = 0.26$), and the stratospheric contribution (slope=$0.52 \pm 0.01$, $R^2 = 0.66$) in the Northern Hemisphere. Removing the seasonality of emissions increases both measurement-model differences and scatter, as we would expect given the seasonality of Northern Hemisphere emissions noted in bottom-up studies (Kirschke et al., 2013). The aseasonal simulation also reduces the offset between TCCON and GEOS-Chem, whereby modeled $X_{CH_4}^t$ and $X_{CH_4}$ are systematically

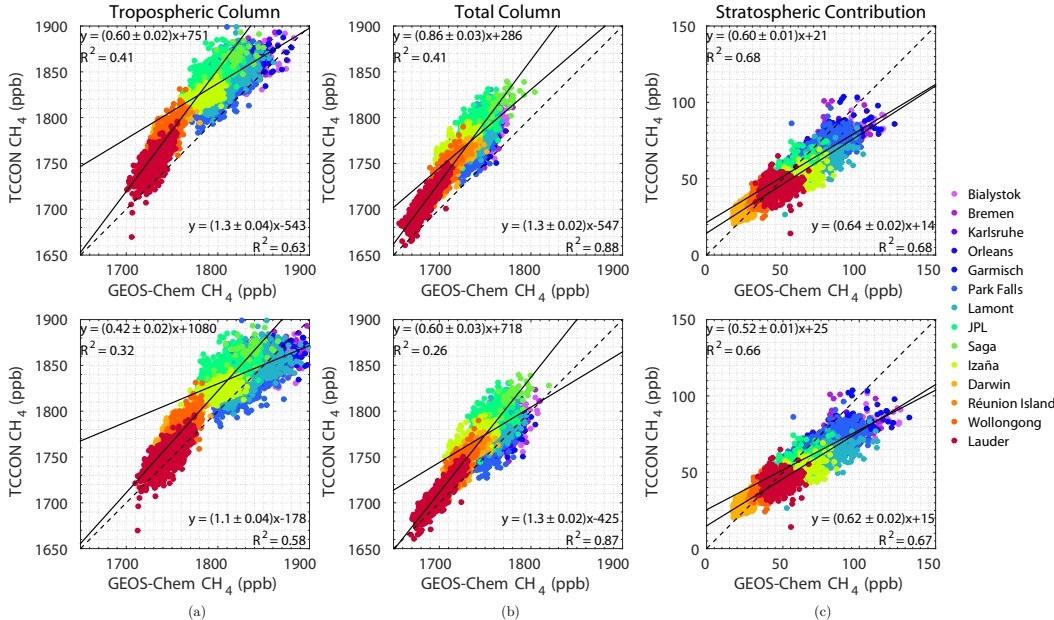

**Figure 4.** Daily median TCCON and smoothed daily mean GEOS-Chem base (top) and aseasonal (bottom) DMFs for (a) $X_{CH_4}^t$, (b) $X_{CH_4}$, and (c) stratospheric contribution. Site colors are as in Fig. 1. Northern Hemisphere least squares regression equations are in the top left, and Southern Hemisphere least squares regression equations are in the bottom right of each plot. Dashed lines mark the one-to-one lines.

low. TransCom-$CH_4$ showed that GEOS-Chem $CH_4$ concentrations tend to be lower than the model median, and much lower than the range of other models when using the same OH fields (Patra et al., 2011). The aseasonal emissions used in this analysis likely reduce this documented imbalance with the model's tropospheric OH fields.

    The $X_{CH_4}$ and $X_{CH_4}^t$ regression equations across Southern Hemisphere sites are nearly equivalent, which suggests that the

Southern Hemisphere is not as impacted by the STE errors as the Northern Hemisphere. This consistency between $X_{CH_4}$ and $X_{CH_4}^t$ could also be a function of the zonal dependence of the stratospheric error: whereas more than half of the Northern Hemisphere sites are north of $45°$N, the most poleward site in the Southern Hemisphere is located at $45°$S. The increased scatter associated with the slightly lower $X_{CH_4}^t$ $R^2$ value of 0.63, compared to the $X_{CH_4}$ $R^2$ value of 0.88, does indicate that the Southern Hemisphere is not exempt from model errors associated with emissions, the OH distribution, or transport. The lower

$X_{CH_4}^t$ slope of the aseasonal simulation (1.1 versus 1.3) illustrates the influence of emissions: removing their seasonality leads to better measurement-model agreement, evidenced by a slope closer to both the one-to-one line and the zero-intercept. We hypothesize that either the seasonality of Southern Hemispheric emissions is too strong or, more likely, errors in the Northern Hemispheric seasonality of emissions drive measurement-model mismatch in the Southern Hemisphere via interhemispheric transport. If this effect was solely due to a changed emissions distribution, we would expect the $X_{CH_4}$ slope to also change for

the Southern Hemisphere sites, if only slightly; instead the slope is equivalent to the base simulation $X_{CH_4}^t$ and $X_{CH_4}$ slopes, and $R^2 = 0.87$, only marginally less than the base simulation $X_{CH_4}$ correlation coefficient.

The stratospheric contribution regression equations differ only slightly between the base and aseasonal simulations: $(0.64 \pm 0.02)x + 14$, $R^2 = 0.68$, versus $(0.62 \pm 0.02)x + 15$, $R^2 = 0.67$. The insensitivity of both the stratospheric contribution and the total columns in the Southern Hemisphere to perturbations in the seasonality of tropospheric emissions could be driven by the smaller vertical gradient across the UTLS that results from the influence of Northern Hemispheric air both in the free

troposphere (Fraser et al., 2011) and the stratosphere (Boering et al., 1995). This effect would also support the interpretation of Northern Hemispheric emissions errors driving disagreement between observations and the model in the Southern Hemisphere.

In the troposphere, $CH_4$ increases from south to north; the stratospheric contribution of $CH_4$, however, increases from the equator to the poles due to the zonal gradient in tropopause height. In the Northern Hemisphere total column, the zonal gradient largely disappears: at high latitudes, the larger tropospheric emissions balances the larger stratospheric contribution.

By contrast, zonal gradients in the Southern Hemisphere troposphere and stratosphere are additive, and greater south to north differences are apparent in the total column.

Figure 5 illustrates how the model differs from ACE-FTS $CH_4$ measurements in the stratosphere over boreal spring (March-April-May) and fall (September-October-November). Excepting above the tropical tropopause, $CH_4$ is considerably lower in the ACE-FTS climatology (v. 2.2, Jones et al., 2012) compared to GEOS-Chem. The difference varies both with altitude and

15 latitude, especially in the Northern spring poleward of 40°N. The vertical gradient is the least pronounced in Lauder, where the stratospheric contributions of TCCON and GEOS-Chem fall most closely to the one-to-one line (Fig. 4).The low $CH_4$ in the tropical mid and upper stratosphere in GEOS-Chem could be a result of too weak vertical ascent to the stratosphere; however, the ACE-FTS data gaps in the tropical troposphere make this hypothesis difficult to test.

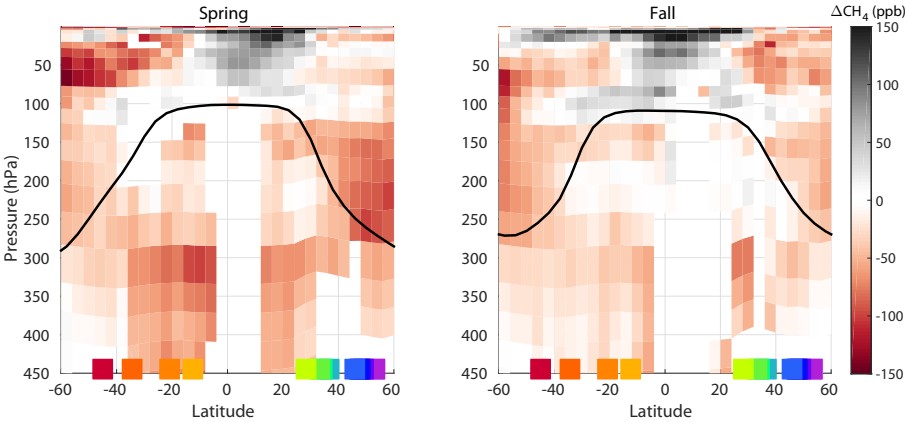

**Figure 5.** Zonally averaged ACE minus GEOS-Chem climatological $CH_4$ mole fractions for boreal spring and fall. Black line represents the mean zonal tropopause level. Site colors of squares on the x-axis are as in Fig. 1.

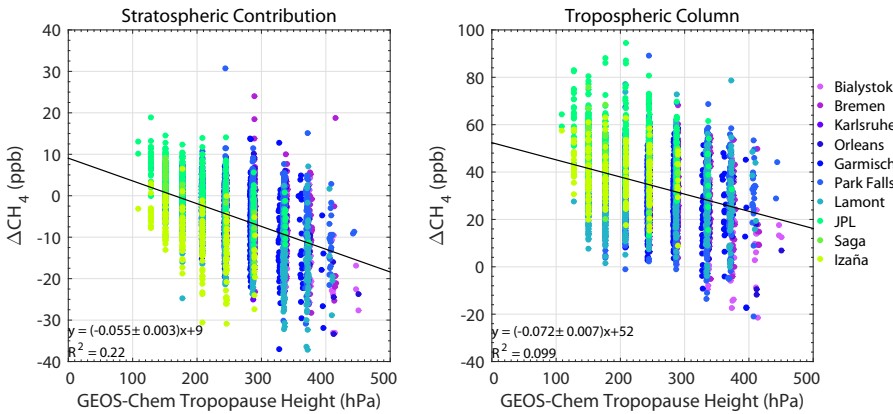

**Figure 6.** TCCON minus GEOS-Chem $CH_4$ column-averaged DMFs as a function of the effective GEOS-Chem tropopause height, shown for Northern Hemisphere sites. Site colors are as in Fig. 1.

## 3.1 Dependence on Tropopause Height

In the Northern Hemisphere, the measurement-model mismatch of the stratospheric contribution increases as the tropopause altitude shifts downward (Fig. 6). As the model's stratospheric portion of the pressure-weighted total column increases, the error in stratospheric $CH_4$ is amplified, causing a larger disagreement with measurements. Because the tropopause height
decreases with latitude, and this gradient increases during winter and spring, this introduces both zonal and seasonal biases. The disagreement exhibits a large spread for relatively few tropopause pressure heights because the model's effective tropopause, that is, the pressure level at which the model divides the troposphere from the stratosphere in GEOS-Chem, is defined at discrete grid level pressure boundaries.

    The tropospheric mismatch ($\Delta X_{CH_4}^t$), by contrast, decreases with tropopause height for the majority of days and exhibits
a much weaker correlation to tropopause height, $0.099$ versus $0.22$ for the stratospheric contribution. Thus, as expected, the tropopause height explains less of the variance in the measurement-model mismatch in $X_{CH_4}^t$: the upper troposphere is generally well-mixed, and chemical loss does not vary with altitude as much as in the lower stratosphere. This weaker relationship also demonstrates that the choice of tropopause used in the tropospheric profile integration does not strongly impact $\Delta X_{CH_4}^t$.

    The relationship between $\Delta X_{CH_4}^t$ and tropopause height has a clear zonal component that indicates that the correlation is
instead a result of another parameter that varies with latitude. The tropospheric slope is dominated by high-latitude sites; the subtropical sites exhibit a much weaker correlation. At Izaña, which is in the sub-tropics at an altitude of 2.4 km, the correlation between $\Delta X_{CH_4}^t$ and tropopause position is weak: the slope of $-0.035 \pm 0.03$ is nearly flat within error, and $R^2$ is $0.025$. By contrast, the stratospheric relationship at Izaña corresponds more closely with the other Northern Hemisphere sites: the slope is $-0.088 \pm 0.02$, and $R^2 = 0.36$.

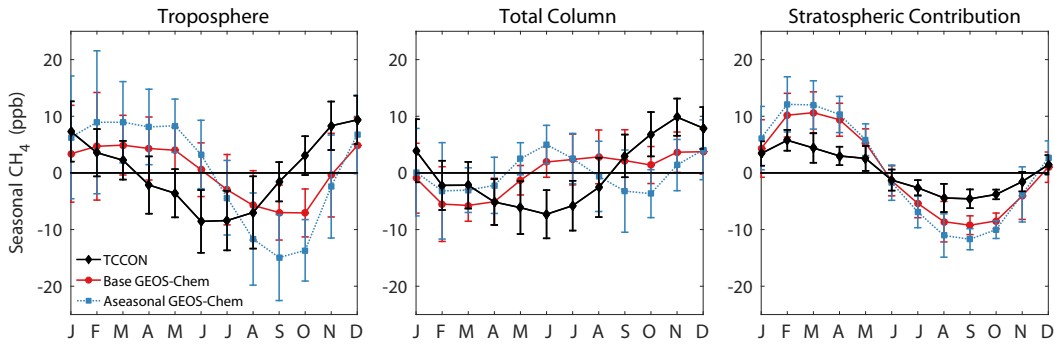

**Figure 7.** Detrended seasonality of TCCON (black diamonds), GEOS-Chem base (red circles), and GEOS-Chem aseasonal (blue squares) $CH_4$ column-averaged DMFs, averaged across Northern Hemisphere sites, except Saga and Réunion Island, which have less than one year of measurements prior to 2012. Error bars denote the $1\sigma$ standard deviation across sites.

## 3.2 Seasonal Agreement

The tropospheric difference between TCCON and GEOS-Chem, $\Delta X_{CH_4}^t$, has a periodic trend indicating that the model error has a strong seasonal component in the troposphere. To isolate stable seasonal patterns from the cumulative influence of emissions, we calculate the detrended seasonal mean column-averaged DMFs for each site. In the Southern Hemisphere, the

measurements and model agree well. Across the Northern Hemisphere sites, however, the seasonality differs (Fig. 7). The seasonal amplitude of GEOS-Chem $X_{CH_4}^t$ is about equal to that of TCCON, but the TCCON $X_{CH_4}^t$ seasonal minimum is in June/July while the GEOS-Chem seasonal minimum is in September/October. Additionally, while TCCON $X_{CH_4}^t$ begins to decrease in January, GEOS-Chem shows some persistence into the spring.

    The seasonal delay also appears in comparisons of GEOS-Chem surface $CH_4$ with National Oceanic and Atmospheric

Administration (NOAA) surface flask measurements at the LEF site in Park Falls (Fig. 8). The seasonality of GEOS-Chem's surface is regulated more by emissions than transport: $CH_4$ peaks in the summer, when wetland emissions are highest (Fig. 10). This contrasts with the flask measurements, which reach a minimum in the summer (Fig. 8). The seasonality covaries remarkably closely with respect to other features: the late winter decrease, spring persistence, and local minimum in October. The spring plateau lasts twice as long as seen in observations, however, and matches $X_{CH_4}^t$, indicating that feature is not the result

of vertical transport between the PBL and free troposphere.

    Not surprisingly, a time lag does not occur in the stratosphere; the TCCON stratospheric seasonal amplitude is less than half but in phase with that of GEOS-Chem (Fig. 7). The vertical inconsistency of the seasonality produces unusual features in the model total column. From January through April, the TCCON and GEOS-Chem $X_{CH_4}$ are consistent because the model's bias in the troposphere is balanced by the larger stratospheric contribution. Starting in May, however, the model diverges from

the measurements as the higher tropopause limits the stratosphere's influence, and the phase lag in the troposphere dominates. This balancing effect is also demonstrated by the greater variance across sites in the model $X_{CH_4}^t$ and stratospheric contribution compared to measurements, but about the same variance in $X_{CH_4}$.

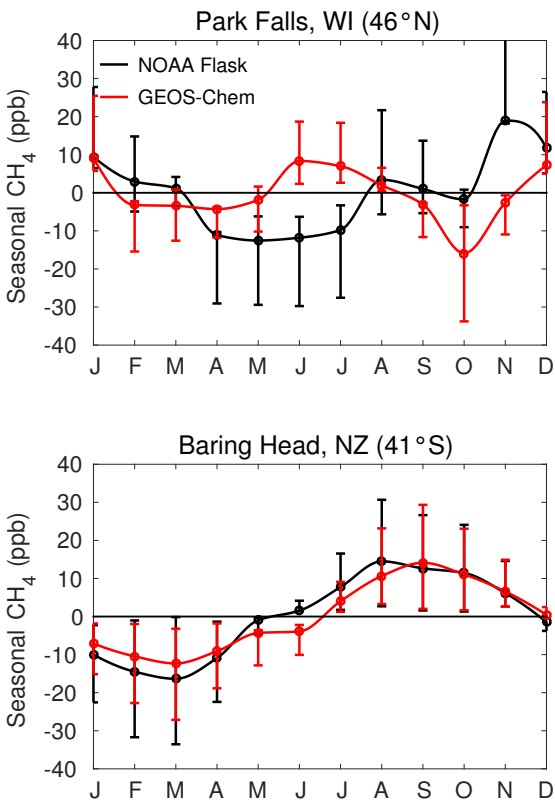

**Figure 8.** NOAA surface flask (black) and GEOS-Chem surface level (red) seasonality of $CH_4$ DMFs over 2005-2011 at Park Falls, WI, USA and Baring Head, NZ. Lower and upper bounds denote the 25th and 75th percentiles, respectively, of detrended data for each month.

For the aseasonal simulation, the tropospheric seasonal cycle amplitude and variance across sites increase (Fig. 7). The greatest model differences, from August through October, are a result of dampening the large wetland fluxes in the base simulation that balance higher OH concentrations. The seasonal amplitude does not increase as drastically in the sub-tropics, where the total emissions are not as impacted by seasonally-varying sources, leading to the greater variance across sites. The second largest difference between simulation amplitudes occurs in the spring, and OH loss could potentially be driving in these months also. The aseasonal simulation spreads the wetland fluxes so as to introduce emissions in the winter and spring, when the OH concentrations are lowest. Another possibility is that the model could be subject to errors that are in phase with the base simulation seasonal emissions, which would then have an ameliorating effect that produces the reasonable seasonal cycle amplitude. The stratospheric contribution does not change, however, further demonstrating that the stratosphere is insensitive to perturbations to Northern Hemisphere emissions.

The impact of a static stratosphere and changing troposphere is to make the seasonality of the aseasonal simulation $X_{CH_4}$ bimodal: the October local minimum in the base simulation becomes a fall absolute minimum. The aseasonal $X_{CH_4}$ agrees with TCCON in late winter, masking the greater disagreement in the troposphere. Notably, the main tropospheric features of the

base simulation, the seasonal phase lag and spring persistence, are still apparent. Thus, the seasonality of emissions prescribed in the forward model is not the driver of the discrepancies between measurement and model $X_{\mathrm{CH_4}}^t$ seasonalities. OH is not likely the driver of these features, as the Northern Hemisphere phase shift also occurs in simulations performed with large changes in OH (Fig. 15, in Appendix B1). Transport is thus the most likely driver of these tropospheric trends in the model.

## 4   Discussion

The stratospheric insensitivity to changes in emissions and tropospheric loss has significant implications for flux inversions. Model inversions use the sensitivity of trace gas concentrations at a given location to perturbations of different emission sources to adjust those emissions so as to match observations at that location. The response of modeled $CH_4$ DMFs to changing emissions depends on the model's transport and chemical loss, as well as assumptions about the seasonal and spatial distribution of emissions relative to each other. Thus the model sensitivity kernel, the linear operator that maps emissions to $CH_4$ concentrations, implicitly includes uncertainties in these terms. The model's stratospheric response to emissions perturbations differ from that of the troposphere and are subject to different transport and loss errors. Because the tropospheric transport errors covary with emissions, they alias into the resulting source attribution.

Comparing measurement and model stratospheric $CH_4$ as a fraction of the total column provides a normalized comparison that isolates differences in the vertical structure from those caused by initial conditions and unbalanced sources and sinks. Figure 9 illustrates the error associated with the normalized stratospheric column and the associated stratospheric contribution to $X_{\mathrm{CH_4}}$ at Park Falls. Although the stratosphere accounts for less than 30% of $X_{\mathrm{CH_4}}$, a relatively small error can produce significant seasonal differences; the springtime error of $4.5 \times 10^{17}$ molec·cm$^{-2}$ (23 ppb) is more than twice the seasonal cycle amplitude. Winter and spring are also when $X_{\mathrm{CH_4}}^t$ is least sensitive to seasonal emissions; by contrast, the error is about 15 ppb in the summer, when seasonal emissions have the greatest influence (Fig. 9, top panel). The seasonality of the stratospheric error will therefore distort the inversion mechanism and thus posterior emissions estimates.

Additional bias is introduced by differences in the seasonal patterns of $\Delta X_{\mathrm{CH_4}}^t$ and $\Delta X_{\mathrm{CH_4}}$. Wetlands are the largest seasonal source of $CH_4$ in models and the largest natural source in flux inventories, and their emissions are very uncertain: estimates range between 142 and 284 TgC·year$^{-1}$ for the 2000-2009 time period (Kirschke et al., 2013). A priori GEOS-Chem $CH_4$ emissions from northern high-latitude wetlands are extremely variable, with large fluxes in June, July and August, moderate fluxes in May and September, and almost no fluxes the remainder of the year (Fig. 10a). Surface $CH_4$ concentrations in models depend on the assumed seasonally-varying emissions. Patra et al. (2011) found that correlations between the seasonal cycles of the forward model averages and in situ observations of $CH_4$ DMFs at the surface varied for a given site by up to $0.78 \pm 0.4$ depending on wetland and biomass burning fields used. Model inversions that scale emissions in a given grid box based on the incorrect seasonality will invariably change the posterior attribution of seasonal emissions. Fraser et al. (2013) found that optimized wetland emissions from inversions that assimilate surface data only are smaller than the priors, while those from inversions that assimilate GOSAT total columns are larger, even if surface measurements are also assimilated. From

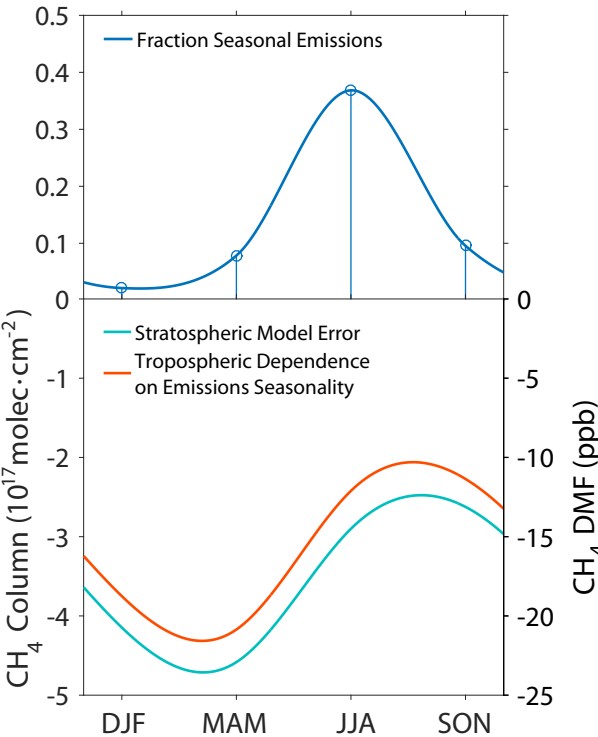

**Figure 9.** Top: Seasonally-averaged fraction of model emissions from seasonally-varying sources, north of $40°$N. Bottom: Seasonally-averaged normalized model stratospheric column error (teal) and the difference between base and aseasonal simulation tropospheric columns (orange) at Park Falls.

this we infer that the transport errors in the model's free troposphere lead to an "optimization" of the prior fluxes of opposite sign to that of the emissions errors that the inversion attempts to correct.

A two to three-month shift in the phase of the $X_{CH_4}^t$ seasonality will produce a strong under- or overestimation of posterior wetland fluxes in late spring through early fall. In an inversion, prior emissions are adjusted in proportion to the deviation of the model's CH$_4$ DMFs from observed values. These posterior emissions are scaled for each sector according to their a priori fraction of total emissions in each grid box. Thus, an increase in posterior emissions relative to the prior in the northern mid and high latitudes during winter will not change emissions from wetlands. For example, Fig. 10b illustrates the sensitivity of posterior wetland emissions to a three-month lag in the Northern Hemisphere (derived by calculating the total emissions resulting from an increase of 1 ppb of CH$_4$ in each tropospheric column and scaling those emissions according to the a priori contribution of wetlands). The tropics and subtropics are less sensitive to a phase shift, but polewards of $40°$N, both the magnitude and seasonality of the difference is significant. Large differences between measured and modeled $X_{CH_4}^t$ are concurrent with low emissions from seasonal sources. The adjustments to prior emissions produced by larger measurement-model disagreement that occur when seasonal sources are a small fraction of total emissions will overestimate posterior emissions from aseasonal

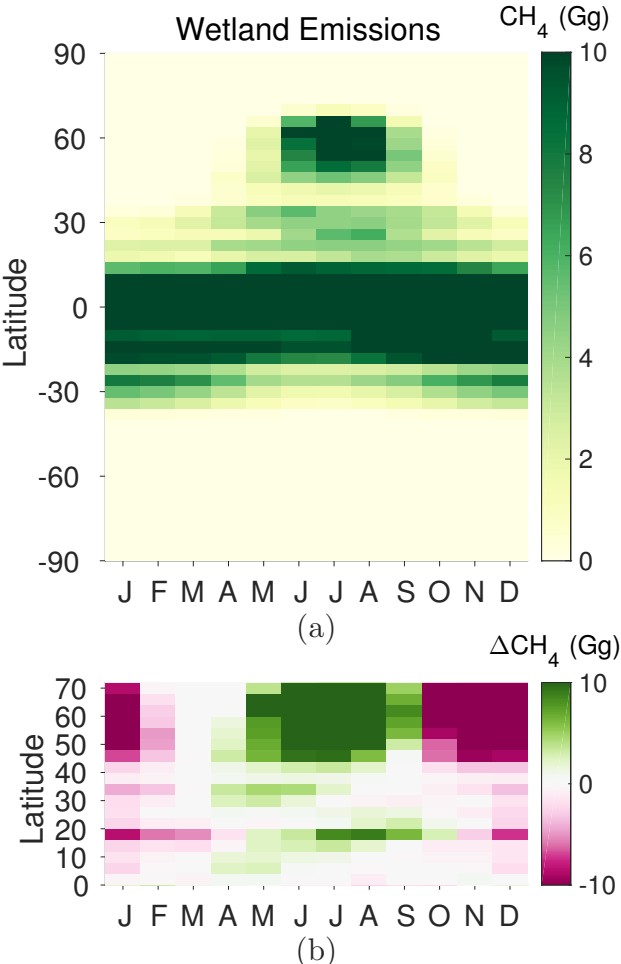

**Figure 10.** (a) GEOS-Chem monthly zonal mean wetland emissions, in Gg. (b) The Northern Hemisphere sensitivity of GEOS-Chem wetland emission attribution caused by a 3-month lag for each 1 ppb increase of $CH_4$ in the tropospheric column, in Gg.

sources. Thus these seasonal errors will bias source apportionment toward emissions that do not vary on timescales shorter than annually.

## 5  Conclusions

Assimilation of total column measurements into CTMs can improve constraints on the global $CH_4$ budget; however, the model's treatment of stratospheric chemistry and dynamics must be carefully considered. This work has compared TCCON and GEOS-Chem pressure-weighted total and tropospheric column-averaged $CH_4$ DMFs, $X_{CH_4}$ and $X_{CH_4}^t$ respectively, parsing out the seasonality of the troposphere and stratosphere and the resulting impacts on $X_{CH_4}$ (Fig. 9a). The Southern Hemisphere measurement-model agreement is robust to changes in emissions or tropospheric OH. In the Northern Hemisphere the model's

stratospheric contribution is larger than that of the measurements, and the mismatch increases as the tropopause altitude decreases. The result is greater model error at high-latitude sites, with the magnitude of this error varying seasonally. Moreover, in the Northern Hemisphere the GEOS-Chem $X_{CH_4}^t$ exhibits a 2-3 month phase lag. The combined tropospheric and stratospheric errors smooth the model $X_{CH_4}$ such that they may agree with total column measurements despite having an incorrect vertical distribution.

Model transport errors coupled with spatial and seasonal measurement sparsity can limit the accuracy of the location and timing of emissions scaling. The differences in the seasonality mismatch across vertical levels amplifies the error uncertainty because the timing of optimized fluxes will be especially susceptible to limitations in model transport. The stronger influence of the stratosphere at higher latitudes due to lower tropopause heights, together with the higher temporal variability of the stratospheric fraction of the total column due to the stronger seasonal cycle of the tropopause, also impacts the seasonality of the meridional gradient of $X_{CH_4}$.

The influence of stratospheric variability on emissions is not unique to the model chosen for this analysis. Bergamaschi et al. (2013) ran TM5-4DVAR inversions using SCIAMACHY column and NOAA surface measurements and found that the mean biases between the optimized $CH_4$ profiles and aircraft measurements differ between the PBL, free troposphere, and UTLS. Seasonal emissions from wetlands and biomass burning vary by $\pm 10$ and $\pm 7$ $TgCH_4$, respectively, from year to year, and the zonal partitioning of posterior emissions is sensitive to the wetland priors chosen. Moreover, the larger changes to emissions and sensitivity to assumptions in the Northern Hemisphere indicate that TM5 is also subject to the strong hemispheric differences found in GEOS-Chem. The TransCom-$CH_4$ model comparison found that the interhemispheric exchange time in GEOS-Chem was near the model median over the 1996-2007 time series (Patra et al., 2011), which suggests that GEOS-Chem's interhemispheric transport, and thus associated errors, is not particularly distinct. Ostler et al. (2015) found that ACTM and other CTMs used in TransCom-$CH_4$ are also subject to transport errors that impact emissions optimization. Furthermore, ACTM profiles show a similar over-estimation of stratospheric $CH_4$, zonally-varying measurement-model mismatch dependent on tropopause height, and a smaller seasonal cycle for Northern Hemisphere $X_{CH_4}$ compared to TCCON.

In this analysis we have used TCCON $X_{CH_4}^t$ derived with the HF-proxy method; however, $X_{CH_4}^t$ calculated using other stratospheric tracers such as nitrous oxide ($N_2O$) (Wang et al., 2014) would provide an additional constraint on models' representations of the stratosphere, as $N_2O$ is not subject to the spectral interference with water vapor that impacts HF. Information about the vertical tropospheric $CH_4$ profile directly retrieved from NDACC spectra (Sepúlveda et al., 2014) can also be used to assess whether transport errors differ at different levels of the free troposphere. Ideally, information from these tropospheric products could be integrated to overcome the limitations of each: the sensitivity of $X_{CH_4}^t$ to prior assumptions of stratospheric-tropospheric exchange and the sensitivity of profile retrievals to UTLS variability (Ostler et al., 2014).

A limitation of the aseasonal simulation was that the distribution of emissions was not identical to that of the base simulation due to the scaling approach we employed. Ideally, the aseasonal emissions for each sector would have been fluxes calculated for each grid box from the base simulation annual emissions. The robustness of the model's tropospheric phase shift that was apparent regardless of the emissions used demonstrates that this feature is not a product of the chosen emissions fields. However, more nuanced analysis on smaller spatial scaled would benefit from simulations that prescribe the annual mean for

each of the seasonal sources. The most recent version of GEOS-Chem has a much more flexible emissions scheme (Keller et al., 2014) that allows these more nuanced experiments to be performed and analyzed.

The insensitivity of model stratospheres to tropospheric change allows for a straightforward solution: prescribed stratospheric $CH_4$ fields based on satellite observations from ACE-FTS, MIPAS (von Clarmann et al., 2009), or a compilation of remote sensing instruments (Buchwitz et al., 2015). As the representation of tropical convection and exchange across the UTLS advances in models and reduce stratospheric isolation, chemical loss, and transport mechanisms would need to be improved. The output from more accurate stratospheric models over the time period of interest could be used to set the stratospheric component in the offline $CH_4$ simulation. For instance, the Universal tropospheric-stratospheric Chemistry eXtension (UCX) mechanism, which has been added to more recent versions of GEOS-Chem, updates the stratospheric component of the standard full chemistry simulation such that $CH_4$ has more sophisticated upwelling, advection and chemical reaction schemes (Eastham et al., 2014). Models that account for interannual variability in both stratospheric and tropospheric dynamics can then assimilate total column measurements to develop more accurate global $CH_4$ budgets.

## Appendix A:  Updates to Tropospheric Methane Data

The TCCON $X_{CH_4}^t$ data used in this analysis were developed as in Saad et al. (2014) with several adjustments to both the parameters used and methodology.

The HF-proxy method for determining $X_{CH_4}^t$ incorporates the relationship between $CH_4$ and HF in the stratosphere, which is calculated using ACE-FTS data. These $CH_4$-HF slopes now use updated ACE-FTS version 3.5 measurements with v.1.1 flags (Boone et al., 2013; Sheese et al., 2015). The data quality flags are provided for profile data on a 1 km vertical grid, which uses a piecewise quadratic method to interpolate from the retrievals (Boone et al., 2013). Additionally, the $CH_4$ and HF measurement errors are now considered in the pressure-weighted linear regression that determines the slopes. All other data processing to produce the $CH_4$-HF slopes followed methods described in Saad et al. (2014). Figure 11 shows the updated annual zonal values used to calculate $\hat{X}_{CH_4}^t$ with Washenfelder et al. (2003) and MkIV (retrieved from http://mark4sun.jpl.nasa.gov/m4data.html) values included for reference (c.f. Saad et al., 2014, Fig. 2). These updates altered $\hat{X}_{CH_4}^t$ for the sites and time period covered in this paper by less than 2 ppb.

The derivation of the tropospheric column in Washenfelder et al. (2003), Saad et al. (2014), and Wang et al. (2014) implicitly assumed that the $CH_4$ profile is continuous across the tropopause; however, the boundary condition for stratospheric $CH_4$ is rather set by tropospheric air transported through the tropical tropopause (Brewer, 1949; Dobson, 1956). Boering et al. (1996) showed that the concentration of $CO_2$ directly above the tropopause can be approximated by introducing a two-month phase lag to the average concentration at northern and southern tropical surface sites: Mauna Loa, Hawaii (MLO) and Tutuila, American Samoa (SMO), respectively. As the $CH_4$ entering the stratosphere originates in both hemispheres (Boering et al., 1995), stratospheric $CH_4$ exhibits a smaller interhemispheric gradient than in the troposphere: about 20ppb, as calculated from ACE-FTS measurements, versus about 50 ppb, taken as the difference at MLO and SMO. To calculate the stratospheric boundary condition for $CH_4$ we remove the seasonal component of the mean of $CH_4$ DMFs at MLO and SMO, which are made

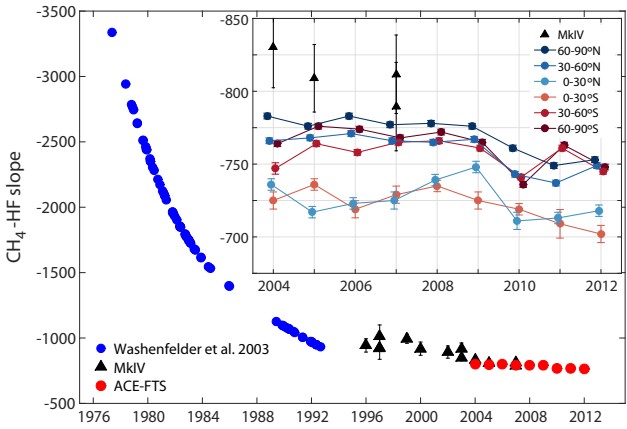

**Figure 11.** Long-term $CH_4$–HF slopes from Washenfelder et al. (2003), MkIV, and updated ACE-FTS measurements. Inset: Time series of zonal pressure-weighted ACE-FTS slopes ($\beta$) used to calculate $\hat{X}_{CH_4}^t$, with error bars denoting the $2\sigma$ standard error. Zonal slopes are offset each year for visual clarity.

available through 2014 by the NOAA Earth System Research Laboratory (ESRL) Global Monitoring Division (Dlugokencky et al., 2016). To capture the interhemispheric gradient observed in ACE stratospheric $CH_4$ measurements, we add and subtract 10 ppb, in the northern and southern extratropics respectively, the limits of which we choose as the Tropic of Cancer ($23°N$) and the Tropic of Capricorn ($23°S$). A constant value is chosen in each hemisphere to reflect the rapid mixing time of air from

the extra-tropics in the region directly above the tropopause, which Boering et al. (1996) found to be less than one month. Within the tropics, we interpolate the boundary condition as a linear function of altitude such that $x_{CH_4}(P^t) = \bar{x}_{CH_4}^s + \frac{10}{23}\lambda$, where $x_{CH_4}(P^t)$ is the boundary condition at the tropopause, $\bar{x}_{CH_4}^s$ is the mean DMF of $CH_4$ at the surface, and $\lambda$ is the latitude of the site.

    Assuming hydrostatic equilibrium, the tropospheric column of $CH_4$, $c_{CH_4}^t$, can be calculated as the integral of the vertical

profile, $x_{CH_4} \equiv x_{CH_4}(P)$, from the surface, $P^s$, to the tropopause, $P^t$:

$$c_{CH_4}^t = \int_{P^t}^{P^s} x_{CH_4} \frac{dP}{gm} = X_{CH_4}^t \frac{P^s - P^t}{g_*^t m} \tag{A1}$$

where $P$ is the pressure height, $g$ is the gravitational acceleration, $g_*^t$ is the pressure-weighted tropospheric value of $g$, and $m$ is the mean molecular mass of $CH_4$ (Washenfelder et al., 2006). The profile of $CH_4$ in the stratosphere can be expressed as

a linear function of pressure altitude, $x_{CH_4}(P) = x_{CH_4}(P^t) + \delta \cdot P$, where $\delta = \frac{dx_{CH_4}}{dP}$ is the stratospheric loss of $CH_4$. This stratospheric loss term is estimated by the HF-proxy method to produce the retrieved tropospheric column-averaged DMF, $\hat{X}_{CH_4}^t$, such that

$$\hat{X}_{CH_4}^t \frac{P^s}{g_* m} = \hat{c}_{CH_4}^t = \int_0^{P^s} x_{CH_4} \frac{dP}{gm} - \int_0^{P^t} \delta \cdot P \frac{dP}{gm} \tag{A2}$$

where $g_*$ is the pressure-weighted column average of $g$. The stratospheric boundary condition can thus be related to the retrieved tropospheric column as

$$\int_0^{P^t} x_{CH_4} \frac{dP}{gm} = \int_0^{P^t} x_{CH_4}(P^t)\frac{dP}{gm} - \hat{c}^t_{CH_4} + \int_0^{P^s} x_{CH_4}\frac{dP}{gm}. \tag{A3}$$

Given the total column integration is the sum of the tropospheric and stratospheric partial columns, and substituting Equation A3:

$$\int_{P^t}^{P^s} x_{CH_4}\frac{dP}{gm} = \int_0^{P^s} x_{CH_4}\frac{dP}{gm} - \int_0^{P^t} x_{CH_4}\frac{dP}{gm} \tag{A4}$$

$$= \int_0^{P^s} x_{CH_4}\frac{dP}{gm} - \int_0^{P^t} x_{CH_4}(P^t)\frac{dP}{gm} + \hat{c}^t_{CH_4} - \int_0^{P^s} x_{CH_4}\frac{dP}{gm} \tag{A5}$$

$$= \hat{c}^t_{CH_4} - \int_0^{P^t} x_{CH_4}(P^t)\frac{dP}{gm} \tag{A6}$$

$$X^t_{CH_4}\frac{P^s - P^t}{g^t_* m} = \hat{X}^t_{CH_4}\frac{P^s}{g_* m} - x_{CH_4}(P^t)\frac{P^t}{g^0_* m} \tag{A7}$$

where $g^0_*$ is the pressure-weighted average of $g$ from the tropopause to the top of the atmosphere. While the molecular mass of air changes as a function of water vapor and thus altitude and gravity changes as a function of both altitude and latitude, assuming constant values of $g$ and $m$ changes $X^t_{CH_4}$ by less than 2 ppb. Thus, to good approximation these variables can be

canceled out:

$$X^t_{CH_4}[P^s - P^t] = \hat{X}^t_{CH_4} \cdot P^s - x_{CH_4}(P^t) \cdot P^t \tag{A8}$$

$$X^t_{CH_4} = \frac{\hat{X}^t_{CH_4} \cdot P^s - x_{CH_4}(P^t) \cdot P^t}{P^s - P^t}. \tag{A9}$$

The surface pressure is measured at each site, and the tropopause pressure is calculated from the TCCON prior temperature

profiles. The uncertainties associated with the interpolated value of the tropopause height are determined by calculating $X^t_{CH_4}$ for $\pm 30\%$ of $P^t$ and adding these confidence intervals in quadrature to the precision error of $\hat{X}^t_{CH_4}$. The aforementioned deseasonalization of $x_{CH_4}(P^t)$ is an approximation that adds another uncertainty. The signal of the tropospheric seasonal cycle of a trace gas entering the stratosphere is apparent directly above the tropopause and both dampens in amplitude and shifts in time with increasing altitude (Mote et al., 1996). Thus, the stratospheric boundary condition is not truly constant throughout

the column, but rather the pressure-weighted sum of these attenuated signals. Calculating $x_{CH_4}(P^t)$ without removing the seasonality, which provides the maximum impact of this uncertainty, decreases $X^t_{CH_4}$ by an average of 1 ppb and 4 ppb in the

Northern and Southern Hemispheres, respectively, and does not alter the seasonal cycle of $X^t_{CH_4}$. Moreover, as described below, the mismatch between the calibrated TCCON $X^t_{CH_4}$ and the in situ aircraft $X^t_{CH_4}$ does not correlate with season ($R^2 = 0.017$). Thus, we retain the simpler computation of deseasonalized $x_{CH_4}(P^t)$ in Equation A9.

Airmass-dependent artifacts were derived for updated values consistently with the total column $CH_4$ (Wunch et al., 2015).

5    Removing these artifacts, the $X^t_{CH_4}$ was then calibrated with in situ aircraft profiles using the same methodology described in Wunch et al. (2010) and including the updates delineated in (Wunch et al., 2015) to produce a calibration correction factor of 0.9700 (Fig. 12). The covariance between the difference between the calibrated TCCON and aircraft $X^t_{CH_4}$ and several parameters were assessed to ensure biases were not introduced into the measurements. These differences had an uncertainty-weighted correlation coefficient of 0.1 for solar zenith angle and uncertainty-weighted correlation coefficients of less that

10    0.02 for tropopause and surface pressures, year, and season. Measurement precisions and errors were determined as in Saad et al. (2014), with the additional uncertainties mentioned in this section included. Individual TCCON sites have median $X^t_{CH_4}$ precisions in the range of 0.1-0.8%, and mean and median precisions are 0.3 and 0.2%, respectively, for all sites through May 2016.

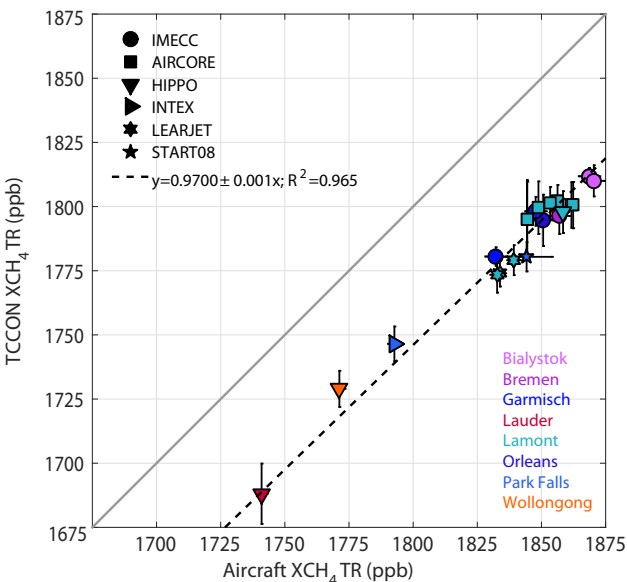

**Figure 12.** Calibration curve of TCCON $X^t_{CH_4}$ (c.f. Wunch et al., 2015, Fig. 8). Site colors are as in Fig. 1. Aircraft campaigns are described in Table 6 of Wunch et al. (2015).

## Appendix B: GEOS-Chem Simulations

### B1 Equilibrium Sensitivity Experiments

All equilibrium runs for a given simulation have identical meteorology, emissions, and OH fields over June 2004-May 2005. Initial conditions for each year are set by the restart files of the previous run. To calculate columns at each site, GEOS-Chem monthly mean mole fractions are adjusted for the monthly medians of the site's daily mean surface pressures and smoothed with the monthly median scaled prior profiles and averaging kernels, interpolated using the monthly median solar zenith angle daily means. Because Park Falls and Lauder are the only TCCON sites that had started taking measurements over this time period, they are the only sites used to generate smoothed columns for the comparisons to the experimental simulations.

Emissions in the aseasonal simulation were derived by running a two-dimensional regression on the annual emissions to determine the scale factors that would produce the smallest residual of total emissions and the interhemispheric gradient. Figure 13 illustrates the difference in total emissions between the base and aseasonal simulations for each zonal band.

The updated OH simulation used OH output from a 2012 GEOS-Chem standard chemistry simulation with extensive updates to the photochemical oxidation mechanisms of biogenic volatile organic compounds (VOCs), described in Bates et al. (2016) and references therein. These were converted to 3D monthly mean OH concentrations to conform to the infrastructure of the GEOS-Chem offline $CH_4$ tropospheric loss mechanism. The OH was then scaled by 90% to keep the lifetime above 8 years, and emissions were scaled by 112% to maintain the same balance between sources and sinks in the base simulation. Figure 14 provides zonal averages of the difference between the base and updated OH columns.

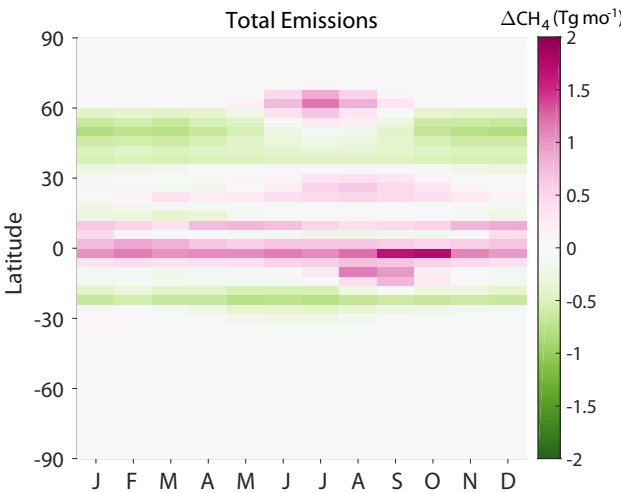

**Figure 13.** Monthly averages of the difference in total $CH_4$ emissions between the base and aseasonal GEOS-Chem simulations, summed over each zonal band, in $Tg \cdot mo^{-1}$.

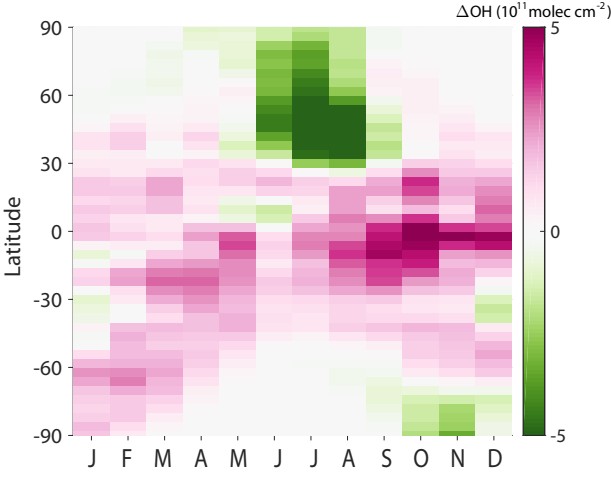

**Figure 14.** Zonal averages of the difference in total column OH (molec·cm$^{-2}$) between the base and updated monthly OH fields.

The full list of simulations run is provided in Table 3, with descriptions and the $CH_4$ emissions, tropospheric OH, and total chemical loss lifetimes. Figure 15 shows each simulation's seasonality of $X_{CH_4}^t$ at Park Falls, with TCCON seasonality plotted as reference, as well as the seasonality of the difference between the base and each simulation.

**Table 3.** List of Sensitivity Experiments

| Run Name | Description | CH$_4$ Lifetime (years) with respect to | | | Final CH$_4$ |
| --- | --- | --- | --- | --- | --- |
| | | Emissions | Tropospheric OH | Total Loss | Burden (Tg) |
| Base | Default OH and Emissions | 9.6 | 10.7 | 9.7 | 4825 |
| Aseasonal | Constant Monthly Emission Rates | 9.6 | 10.7 | 9.7 | 4872 |
| Updated OH | Monthly OH fields from Standard Chemistry + Biogenic VOCs, scaled down by 10% | 8.5 | 9.4 | 8.6 | 4828 |
| Unscaled Updated OH | Monthly OH fields from Standard Chemistry + Biogenic VOCs | 7.7 | 8.4 | 7.8 | 4917 |
| 90% OH | Default OH scaled down by 10% | 10.5 | 11.9 | 10.7 | 5296 |
| 110% OH | Default OH scaled up by 10% | 8.8 | 9.7 | 8.8 | 4425 |
| Scaled Rice Emissions | Rice Emissions Increased by 20% | 9.6 | 10.7 | 9.6 | 4780 |
| No Wetlands | Wetland Emissions Turned Off | 10.7 | 10.6 | 9.5 | 3768 |
| Scaled Livestock Emissions | Scale livestock emissions by 50% | 9.6 | 10.7 | 9.6 | 4359 |
| MERRA | MERRA meteorology fields | 9.6 | 10.7 | 9.6 | 4849 |
| Tropopause Level | Set top of troposphere 2 vertical levels higher | 9.6 | 10.6 | 9.6 | 4855 |

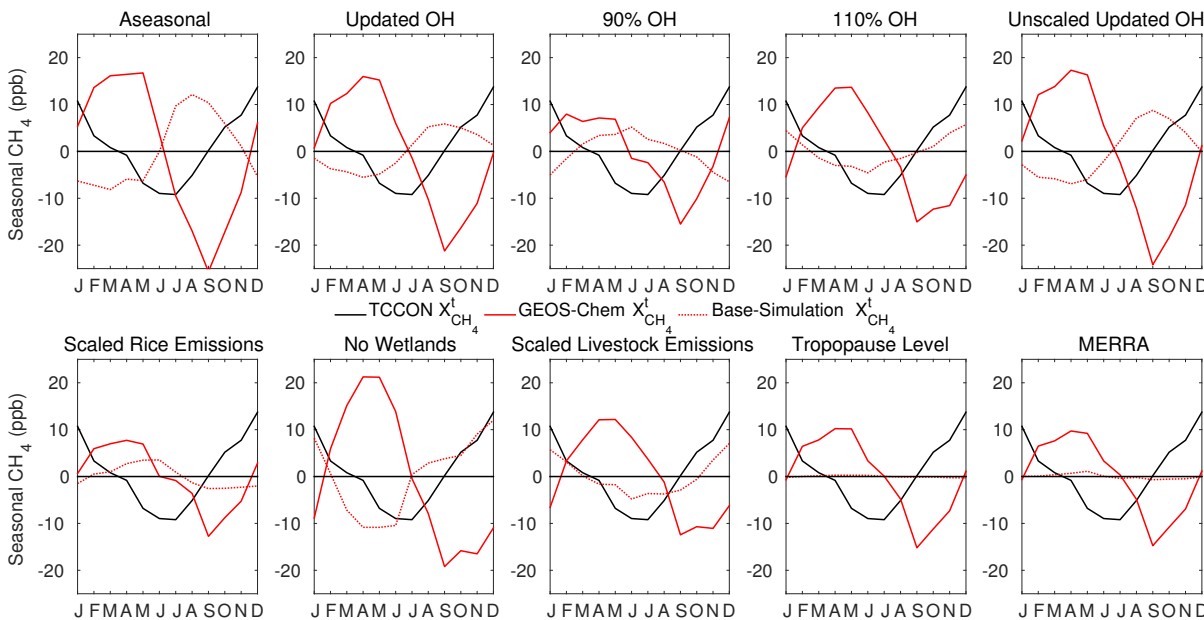

**Figure 15.** Seasonality of tropospheric methane ($X_{CH_4}^t$) at Park Falls for TCCON (black solid line), GEOS-Chem (red solid line), and the difference from the base simulation (dotted red line) for each of the sensitivity experiments, in ppb.

## B2 Derivation of Dry Gas Values

Versions of GEOS-Chem prior to v.10 have inconsistencies in wet versus dry definitions of pressure, temperature, and air mass, which propagate into model diagnostics and conversions calculated using these terms. As a consequence, $CH_4$ concentrations are output assuming air masses that include water vapor but calculated with the molar mass of dry air. For all comparisons in this analysis $CH_4$ DMFs are calculated taking into account the GEOS-5 specific humidity, SPHU (in units of $g_{H2O} \cdot kg_{air}^{-1}$), such that

$$x_{CH_4,dry} = \frac{x_{CH_4}}{1 - SPHU \times 10^{-3}} \tag{B1}$$

where $x_{CH_4}$ is the model profile in mole fractions. Dry air profiles were derived by subtracting the water vapor mole fraction, also calculated from the GEOS-5 specific humidity, from the total air mass at each pressure level, as in Wunch et al. (2010); Geibel et al. (2012).

## B3 Model Smoothing for Measurement Comparisons

Base and aseasonal daily runs were initialized using $CH_4$ fields from their respective 34th equilibrium cycles. Daily $CH_4$ mole fractions averaged over both 24-hour and 10-14 local time were output to test whether TCCON's daytime-only observations would introduce a bias in the comparisons. Measurement-model differences were not sensitive to averaging times. Comparison of measurements to model columns produced using the 24-hour and 10-14 local time averages produce equivalent slopes and

only slightly different intercepts and correlation coefficients. The seasonality of 10-14 local time column-averaged DMFs does not differ, except that the fall seasonal maximum of the adjusted troposphere and stratospheric contribution at Park Falls in October, one month later than the 24-hour column-averaged DMF seasonality.

CH$_4$ dry vertical profiles for each grid box associated with a TCCON site, $\boldsymbol{x}^m_{\mathbf{CH_4}}$, were smoothed with corresponding FTS column averaging kernels, $\boldsymbol{a}_{\mathbf{CH_4}}$, and scaled priors for each day and vertically integrated using pressure-weighted levels:

$$X^s_{\mathrm{CH_4}} = \gamma_{\mathrm{CH_4}} \cdot X^a_{\mathrm{CH_4}} + \boldsymbol{a}^{\S}_{\mathbf{CH_4}}(\boldsymbol{x}^m_{\mathbf{CH_4}} - \gamma_{\mathrm{CH_4}}\boldsymbol{x}^a_{\mathbf{CH_4}}) \tag{B2}$$

where $X^s_{\mathrm{CH_4}}$ is the smoothed GEOS-Chem column-averaged DMF, $\gamma_{\mathrm{CH_4}}$ is the TCCON daily median retrieved profile scaling factor, and $\boldsymbol{x}^a_{\mathbf{CH_4}}$ and $X^a_{\mathrm{CH_4}}$ are respectively the a priori profile and column-integrated CH$_4$ DMFs (Rodgers and Connor, 2003). The pressure weighting function, $\boldsymbol{h}$, was applied such that $X = \boldsymbol{h}^T\boldsymbol{x}$. TCCON priors were interpolated to the GEOS-Chem pressure grid, and GEOS-Chem pressure and corresponding gas profiles were adjusted using daily mean surface pressures local to each site (Wunch et al., 2010; Messerschmidt et al., 2011). The averaging kernels were interpolated for the local daily mean solar zenith angle and the GEOS-Chem pressure grid so that it could be applied to the difference between the GEOS-Chem and TCCON profiles as $\boldsymbol{a}^{\S}\boldsymbol{x} = \sum_{i=1}^{N} a_i h_i x_i$ from the surface to the highest level, $N$, at $i$ pressure levels (Connor et al., 2008; Wunch et al., 2011b). Figure 16 shows how the smoothed column compares to the column that only uses the dry gas correction.

## Appendix C: Derivation of Stratospheric Contribution

Considering the CH$_4$ profile integration as in Equation A4, and substituting the profile of CH$_4$ in the stratosphere, $x_{\mathrm{CH_4}}(P) = x_{\mathrm{CH_4}}(P^t) + \delta \cdot P$, described in Appendix A, the total column is calculated as:

$$\int_0^{P^s} x_{\mathrm{CH_4}}\frac{dP}{gm} = \int_{P^t}^{P^s} x_{\mathrm{CH_4}}\frac{dP}{gm} + \int_0^{P^t} [x_{\mathrm{CH_4}}(P^t) + \delta \cdot P]\frac{dP}{gm} \tag{C1}$$

$$X_{\mathrm{CH_4}} \cdot P^s = X^t_{\mathrm{CH_4}}[P^s - P^t] + x_{\mathrm{CH_4}}(P^t) \cdot P^t + c^{\delta}_{\mathrm{CH_4}} \tag{C2}$$

where $c^{\delta}_{\mathrm{CH_4}}$, is the pressure-weighted column average of CH$_4$ loss in the stratosphere. Rearranging terms, Equation C2 becomes:

$$[X_{\mathrm{CH_4}} - X^t_{\mathrm{CH_4}}]P^s = [x_{\mathrm{CH_4}}(P^t) - X^t_{\mathrm{CH_4}}]P^t + c^{\delta}_{\mathrm{CH_4}} \tag{C3}$$

$$X^t_{\mathrm{CH_4}} - X_{\mathrm{CH_4}} = [X^t_{\mathrm{CH_4}} - x_{\mathrm{CH_4}}(P^t)]\frac{P^t}{P^s} - \frac{c^{\delta}_{\mathrm{CH_4}}}{P^s} \tag{C4}$$

such that the difference between the tropospheric and total column-averaged DMFs is a function of the two terms governing the stratospheric contribution to the total column: the gradient across the tropopause, $x_{\mathrm{CH_4}}(P^t) - X^t_{\mathrm{CH_4}}$, and stratospheric CH$_4$ loss, $c^{\delta}_{\mathrm{CH_4}}$. The stratospheric contribution is thus a proxy for the impact of stratospheric variability on the total column of CH$_4$: given a constant tropospheric column, as the stratospheric contribution becomes larger the total column-averaged DMF becomes smaller.

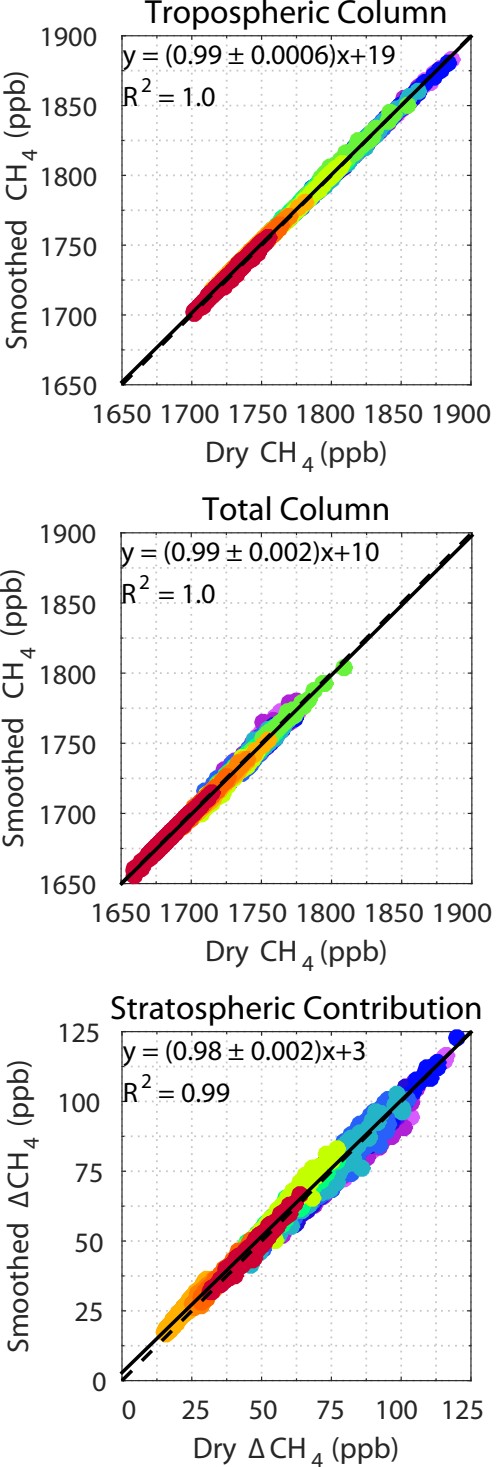

**Figure 16.** GEOS-Chem smoothed versus dry integrated $CH_4$ DMFs for base simulation tropospheric columns, total columns, and stratospheric contribution. Site colors are as in Fig. 1. Dashed lines mark the one-to-one lines.

*Acknowledgements.* This work was supported by NASA Headquarters under the NASA Earth and Space Science Fellowship Program grant NNX14AL30H and NASA's Carbon Cycle Science program. Park Falls, Lamont, and JPL are funded by NASA grants NNX14AI60G, NNX11AG01G, NAG5-12247, NNG05-GD07G, and NASA Orbiting Carbon Observatory Program; we are grateful to the DOE ARM program and Jeff Ayers for their technical support in Lamont and Park Falls, respectively. Darwin and Wollongong are funded by NASA grants NAG5-12247 and NNG05-GD07G and the Australian Research Council grants DP140101552, DP110103118, DP0879468 and LP0562346, and Nicholas Deutscher is supported by an Australian Research Council Fellowship, DE140100178; we are grateful to the DOE ARM program for technical support in Darwin. Bremen, Bialystok, and Orleans are funded by the EU projects InGOS and ICOS-INWIRE and by the Senate of Bremen. Réunion Island is funded by the EU FP7 project ICOS-INWIRE, the national Belgian support to ICOS and the AGACC-II project (Science for Sustainable Development Program), the Université de la Réunion, and the French regional and national organizations (INSU, CNRS). From 2004 to 2011 the Lauder TCCON program was funded by the New Zealand Foundation of Research Science and Technology contracts CO1X0204, CO1X0703 and CO1X0406. We thank Shuji Kawakami for his technical support in Saga. We thank Peter Bernath, Kaley Walker, and Chris Boone for their guidance using the ACE-FTS data, which were obtained through the Atmospheric Chemistry Experiment (ACE) mission, primarily funded by the Canadian Space Agency. We are grateful to Geoff Toon for his continuous efforts developing the GGG software, for providing the MkIV data, and his input on the manuscript. We thank Arlyn Andrews for providing the LEF surface flask data, which were generated by NOAA-ESRL, Carbon Cycle Greenhouse Gases Group. Baring Head NIWA surface data were provided courtesy of Gordon Brailsford, Dave Lowe and Ross Martin. We would also acknowledge the contributions of in situ vertical profiles from the AirCore, HIPPO, IMECC, INTEX, Learjet, and START08 campaigns. We are grateful to Kelvin Bates for providing monthly OH fields for the GEOS-Chem Updated OH sensitivity experiments.

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
