# Peer review of "Seasonal Variability of Stratospheric Methane: Implications for Constraining Tropospheric Methane Budgets Using Total Column Observations"

_Atmospheric Chemistry and Physics, 2016_

## Referee Comment (RC1) · Anonymous Referee #2 · 7 Jun 2016

General comments

This paper by Saad et al., compares the agreement between Total Carbon Column Observing Network (TCCON) and a chemistry transport model (GEOS-Chem) for total and tropospheric column-averaged mole fractions of methane, the second anthropogenic greenhouse gas. From this comparison they infer possible consequences on the methane emissions as estimated by atmospheric inversions using chemistry transport models. The main results include the largest discrepancies between model and observations in the Northern hemisphere stratosphere increasing with latitude as the

tropopause height decreases, and a lag in the model's tropospheric seasonality most probably driven by transport errors. One interesting results is that these errors partly compensate in the total column of methane indicating the possibility to get reasonable agreement for total columns with a wrong vertical transport. This work has implications for atmospheric inversions although the precise quantification of the impacts of the errors found in this paper remain partly to be done. It addresses an important matter as transport model errors are the 2-nd largest cause of uncertainty, after observations space and time density, in atmospheric inversions. Many papers have addressed impact of pbl (rectifier effect) or large-scale horizontal transport (e.g. inter-hemispheric exchange time) but less the impact of vertical transport (e.g. Locatelli et al. 2015).

Nevertheless, the paper needs attention before publication in ACP. It lacks precision in the text in many places (see specific comments), so are legends of some figures. Several important sentences, often when synthetizing results are confusing and not clear and make the reading not fluid at all with this version (see specific comments). I find the result section, a bit too descriptive, not providing systematically explanations or hypotheses for the inferred results. This has to be improved as it is not done either in the discussion part. About the hypotheses, for instance among several other things reported below, I wonder why the aseasonal run disable the seasonal emissions and scale up the rest instead of prescribing the annual mean of seasonally changing sources ? This is strange as it changes the spatial distribution of emissions on the top of the suppression of seasonality. Also, the implication for atmospheric inversions should be more clearly expressed in the discussion section.

Specific comments

Abstract : " large number of highly variable sources Âż not all methane source are highly variable. On what scale ? And sinks ? I suggest because of a large number of uncerain sources and sinks.

Page 2 : lines 1-5 : the words Âń atmospheric inversion Âż should appear somewhere

in this paragraph. Lines 16-17:Do they have the same bias as aircraft observations of clear-sky only measurements (aircraft do not fly in bad weather conditions)? It is worth noticing this issue somewhere. Lines18-20 : Fraser et al : how did they do that ? did they account for observation systematic errors as well ? Please be more precise when quoting papers. Idem for Wecht et al. Lines 33-35 : ambiguous sentence. Please rephrase. Indeed tropospheric CTM do not reproduce well stratospheric transport. . .

Page 3 : Line 6 : Âń systematic model biases Âż : strange expression. Maybe systematic errors would be enough. What about the random part or errors? Do you address this as well ? Please reformulate. Line 6 : "seasonal cycle and spatial distribution of CH4 Âż concentrations ? emoissions ? please be more precise. Line 15 : it would be good to briefly recall how the TCCON total columns are inferred. In particular, what is the influence of the modelled CH4 profile used in the retrieval (as a prior) on the final product. As this profile comes from a model, it would be worth commenting on this considering the topic of the paper. L16 : Âń precise Âż : please be more quantitative here or remove the word. How precise compared to surface networks for instance? how is your data uncertainty estimated ?

Page 4 : Lines 14-15 : please provide a reference for emissions and OH. Do they vary inter-annually ? For OH concentrations, what is your ratio NH/SH ? More precisions are needed here. Indeed you release emissions evey hour but their time evolution is monthly or annually probably. Please precise this not to le the reader think that we know methane emissions with an hourly time step !

Section 2.2 : It would be useful to position GEOS-CHEM with other transport models based on previous Transcom-like experiment (e.g. : Patra et al., 2011): is it a "fast" model ( inter-hemispheric exchange time ?), what about stratosphere/troposphere exchange time ? . . . It would be very useful for other modellers to use the results of the paper.

Page 5 : Line 1-2 : this first sentence needs precision : what is GGG2014 ? What

is GEOS5 ? Acronyms have to be defined and explained Lines 10-15: the choice to disable the seasonal emissions and scale up the rest is strange as it changes the spatial distribution of emissions on the top of the suppression of seasonality. Why not prescribing the annual mean of seasonally changing sources ? Line 15 . What is "TCCON daily median scaled priors Âż ? you need to provide more details here. What is the influence of these Âń priors Âż on the TCCON products and on the comparison proposed here. Line 20 : Âń While XtCH4 20 changed slightly Âż : how much is the change ? please provide % for instance. Why only testing above levels ? please provide explanations. Line 24: "small". Please be more precise. Remain within ±5 ppb for instance ? Idem for larger NH changes : 'varies from -10 to +13 ppb ?

Page 6 : Lines1-2 : what do you mean by Âń common Âż ? Why the age of air increases when seasonality is supressed ? Please provide more clear explanations. Line 3 : "relatively short Âż : please provide an estimate

Page 7 : Line 9-10 : tropospheric slope does not seem lower than one for southern stations. Indeed it seems there is a little north-south gradient in the tropospheric slopes. Did you investigate it ?

Figs 4 : this figure is not enough analysed. You do not comment : - the negative bias of GEOS-CHEM at most sites for the trospospheric & total columns (4ab) - the fact that stratospheric columns of GEOS-CHEM seems underestimated for more southern sites and overestimated for more northern sites (4c) - possible reasons for the poorer agreement in the stratosphere. You may also consider two slopes, one for the southern stations (larger thabn 1) and one for the northern stations (smaller than 1) on fig 4a, or a non linear continuous decrease of the slope from south to north. Why only keeping a global slope ?

Page 8 Line 3-6 : any possible explanation for the differences with ACE ? Line 12-14 : "As the effective..pressure heights" : unclear sentence. Please rephrase. Page 10 line 10 : Âń production Âż or emissions ?

Page8-9 Line 15-4 : the part about troposphere is confusing as figure 6b shows similar trend for stratosphere and troposphere but you mention in the text much lower sensitivity. Please clarify this section.

Page 10 Line 5 : "production" do you mean emissions as there is no methane 3D production in the atmosphere ? More, your statement brings more the summer large wetland emissions as an explanation for the phase of the modelled signal than the loss which should produce more a fall maximum as in surface observations (although Par falls is not the best example to discuss seasonal variations as the signal is complex). Please clarify.

Page 11 : Lines 1-3 : please develop a bit why you discard OH as an hypothesis to explain the inferred changes? Lines 7-8 : "The model sensitivity kernel implicitly includes .. Âż well do you mean variance matrices associated with observations ? with prior emissions ? Indeed, transport errors are generally implicitly include in atmospheric inversions by inflating observations errors but are not part formally of the variance matrix of emissions. Lines 7-8 : "which are compounded if vertical levels are subject to different errorsÂż Confusing sentence. What do you mean ? pleas clarify.

Page 12 : Line 6 'Although the stratosphere accounts for about 30% ' if you refer to top panel of figure 9, I suggest up to 35 % (JJA) Fig9 : The legend of figure 9 is unclear. Top panel : fraction of what ? Bottom panel : the orange curve is a difference or the error of the aseasonal ? Unclear. line 10 : "The seasonality of the stratospheric error will therefore distort the inversion mechanism and thus posterior emissions estimates. Âż : well only if these error are not included in the inversion variance matrices. I would be more confortable writing may distort or precise the conditions of influence of the seasonality in the stratospheric signals on surface emissions through inversions. Line 11 : Âń product of transport errors Âż : how did you evaluate the possibility of issues related to OH radicals ? Lines 10-12 : it is never mention except in caption of figure 4 that 't' in CH4t refers to troposphere. "their emissions are very uncertain Âż : you may quote a recent estimate such as in Kirschke et al. 2013 or IPCC.

Page 13 : Lines 7-8 : "both the magnitude and seasonality of the difference is significant Âż : the unit (tons) makes it difficult to say so. There is obviously a sensitivity if transport error shift the seasonality but what does it give in terms of ppb ? or in terms of % of initial emissions ? This would be more clear for the reader. Lines 8-10 : "The largest disagreements between measured and modeled Xt occur . . . than annually. Âż This sentence is unclear to me. Please rephrase.

Page 14 : line16 "the meridional gradient Âż of what ? emissions ? concentations ? Unit of figure 12 ? Kg/yr ? Maybe change to Tg/gridbox or Tg/yr/°latitude ?

Conclusions Line 3 : re-precise in the start of conclusion the you used GEOS-CHEM and what are XCH4 and XtCH4 as it has to be readable by itself.

Page 15, lines 1-5 : If stratospheric ch4 is largely independent from tropospheric CH4, is it worth developing full tropospheric and stratospheric chemistry models or prescribing stratospheric CH4 based on satellite observations is enough ?

---

## Referee Comment (RC2) · Anonymous Referee #3 · 11 Jun 2016

The manuscript presents a study looking at forward simulations of methane in the GEOS-Chem model, and compares the column-integrated tropospheric/stratospheric column in the model with that derived from tropospheric/stratospheric retrieval of $XCH_4$ from TCCON employing the HF method outlined in Saad et al. (AMT, 2014). The idea, as outlined in the abstract, is compelling: models might match the total column methane mixing ratio measurements of TCCON (and, by extension, satellites), while having compensating errors in the tropospheric and stratospheric components, which would, in turn, bias flux inversions based upon such a model. However it is not clear to me that this study demonstrates this.

[Figure]

My first major concern is that the fluxes used for the "Base" case do not actually match the total column TCCON measurements all that well. This can be seen somewhat by the top row of scatter plots in Figure 4. The correlation between the the total column simulated by GEOS-Chem has a correlation with the TCCON measurements of 0.86, which is even a bit lower than the correlation of the tropospheric columns, which are arguably more relevant for flux inversions. But more worrying, in Figure 7 it can be seen that the seasonality of the total column across the TCCON northern hemisphere sites considered is completely wrong. This inability to capture the seasonal cycle in the total column means that only limited conclusions can be drawn from assessing the (slightly different) mismatch in the two parts of the column. Thus I think the main weakness of this paper is the choice of fluxes used for the forward simulation. These fluxes are only listed in terms of categories, with no itemization of which anthropogenic inventory (I guess EDGAR4.X?), which "other natural emissions", or which model was used for the very important seasonal wetland and rice fluxes. At very least this needs to be amended and clarified. It's fine that the fluxes are added to the model at 60 second increments, but I guess that aside from fires and wetlands/rice the fluxes are constant throughout the year? Or did you employ a diurnal or weekly or annual cycle? And what about the OH fields? Is there a reference for where these came from? Have they been optimized via methyl chloroform or similar?

What would have been a more relevant choice for this type of study would be to use optimized fluxes, resulting from an atmospheric inversion using the same model. There are a few groups working on methane inversions with GEOS-Chem, so such fluxes should not have been difficult to find through collaboration. Then you would have been able to start with a seasonal cycle in the column that is actually consistent at the TCCON sites, assuming that the TCCON sites were assimilated in the inversion. This would have made the analysis more relevant, and it would be my strongest recommendation for improving this study. Without this, I am not sure that the conclusions are clear enough to warrant publication in ACP.

Another concern related to the choice of fluxes relates to the method used for the aseasonal simulation. The manuscript describes that the seasonal fluxes (fires, wetlands, and rice) were "disabled" (I assume this means set to zero?), and then the other fluxes were scaled up to maintain the fluxes and the approximate (but certainly not exact, as showns in Figure 11) latitudinal distribution. Why not simply use an annual mean of the variable fluxes? Then you are not changing two things at once (geographic distribution and temporal variability) and attempting to attribute the changes to only one of the factors.

My next major concern is related to the numerics of how the stratospheric and tropospheric model columns are divided. I do not understand how the the statospheric column-integrated dry air mole fractions have values around 30-100 ppb (from Figure 4). This seems very, very low. Looking at the prior profiles from Wunch et al. (2011), Figure 2, the stratospheric values of $CH_4$ range from 500-1800 ppm. I am not sure if this can be explained by the weighting with the pressure-weighted averaging kernel, as the methane column averaging kernel is actually rather flat (from Figure 4, Wunch et al., 2011). Also from Figure 1 of Saad et al. (2014), the only mixing ratios of stratospheric methane less than even 500 ppb seem to be over 40 km or so, which is far above even the highest tropopause. I had postulated that perhaps you had calculated the mixing ratio not in parts per million molecules of stratospheric air but rather of total column air (in which case it should have been explained). Although I would not advocate for such an approach, in that case the stratospheric partial column dry air mole fraction could be added directly to the tropospheric dry air mole fraction to get the total column dry air mole fraction. Looking again at Figure 4, this is clearly not the case: the tropospheric column is clearly larger than the total column. This needs to be clarified.

One other concern was the consistency of the model tropopause with that from the TCCON retrievals. You mention testing the impact of moving the tropopause model layer up one or two levels, but this does not allow for potential seasonal or regional variability in the match between the two. At very least the (latitude- and seasonal-
dependent) correlation between the model and retrieval tropopause heights should be presented in some way.

More minor comments:

P3, second paragraph: This sounds like you're describing atmospheric inversion while going out of your way not to call it "inversion". Or are you referring to optimization only by processed-based scaling of set spatial fields? Please clarify, and if you mean inversion, please say so.

P2, L10: The reference to Stephens et al. (2007) here seems not to fit so well - this study was looking at aircraft profiles rather than column-integrated information.

P4, L14: Although I mentioned it already above, there needs to be some citations to describe the model and fluxes used.

P5, L9-10: In Appendix A1 I coudln't find any real description of the OH sensitivity runs. Do your OH fields have seasonality? This experiment is insufficiently described.

P5, L15: I was a bit confused here: are the means and medians for all values over the day, over just over those where TCCON measurements were made?

In general I found the use of "DMF" to mean "column-integrated dry air mole fraction" to be rather confusing. Flask measurements also measure dry air mole fraction, so DMF on its own does not tell the reader that an integrated column is being discussed. This is found throughout the manuscript and should be clarified.

Figure 3: The caption says that the stratosphere shows a seasonal cycle of 15 ppb at Park Falls, but in the figure looks like more like 30 ppb. Please explain. I was also surprised to see that Park Falls appears to have a larger seasonal cycle in hte stratosphere than in the troposphere for the Base case. This doesn't make sense to me. Please explain.

P7, L13-14: What about the significant figures on the slopes (e.g. 1.1 $\pm$ 0.020).

P8, L5-7: I'm not sure that Figure 4 shows a good agreement between the stratospheric columns of TCCON and GEOS-Chem. Yes, the clump of points is closer to the 1:1 line, but it hardly forms a line at all. Is the correlation coefficient for this one station really notably higher?

Figure 5: Again I'm confused about the calculation of the stratopsheric column. For instance, we can see from Figure 4 that the stratospheric column simulated by GEOS-Chem is around 50 ppb. Then looking at Figure 5, ACE-FTS minus GEOS-Chem seems to show a difference of approximately -50 ppb around 45 degrees south. Does this mean that the ACE-FTS measurements are showing close to zero methane? In general there seems to be better agreement between TCCON and GEOS-Chem (Figure 4) than ACE-FTS and GEOS-Chem (Figure 5), but it is difficult to tell from the figures presented. Could you comment on this? How do ACE-FTS and TCCON agree?

Figure 7: I am very surprised to see that the aseasonal simulations have higher seasonal cycles in both the stratosphere and the stratosphere than the base case. Are you sure of this result? What role does the (potential) seasonality of the OH sink have here? And what about the sampling throughout the year? Are there enough measurements at Bremen in December and January, or is part of this seasonality a question of shifting sampling throughout the year? Related to this: I assume you are only considering days on which there are TCCON measurements in the model analysis? Another surprise here is that that seasonal cycle of the tropospheric and stratospheric columns in the aseasonal case are essentially in phase, yet when the total column is considered, a bimodal seasonal cycle is found. How can this be?

P10, L11: I disagree with this statement: it seems that the seasonal cycle of the modelled stratospheric columns precede the seasonal cycle of TCCON by a good month.

Figure 8: The smoothing carried out here is not informative. Why not a box and whiskers plot to show how variable the data really are? Also, Park Falls is rather a tricky station with quite a lot of local influence and not a clear seasonal cycle. Perhaps

[Figure]

another station would be more informative? Also, is the temporal sampling of the model consistent with that of the rather sparse flasks? In a broader sense I'm not sure what the real message here is. We see already in Figure 7 that the GEOS-Chem run does a very poor job of representing the seasonal cycle in the NH column: would you expect it to be better at the surface?

Figure 9: Please label the plots (especially upper panel).

Figure 10: The y-axis should have the same scale for the top and bottom figures, even if only part of the range is shown. I was also not quite sure about the units here. $10^6$ kg is 0.001 Tg, so the bright yellow (10 $10^6$ kg $CH_4$) is 0.01 Tg $CH_4$. But then in Figure 11 the increments between the seasonal and aseasonal run seem to be rather on the order of 1 Tg $CH_4$ mo$^{-1}$, which is two orders of magnitude higher. Or have I missed something here?

P13, L5-6: I did not quite understand the description of what you did here. You write "derived by calculating the total emissions resulting from an increase of 1 ppb of $CH_4$ in each surface grid box". Do you mean by calculating the emissions required to cause a 1 ppb increase in each surface grid box? How often were you adding this increment? Monthly? Do you consider the effect that these emissions have on the concentrations of neibouring grid boxes? Is there a reference that explains this procedure in a bit more detail? Based on what is written here, I could not reproduce the experiment.

P14, L8: I don't think you have convincingly shown that the seasonal lag is a function of transport, and not, say, your sink, or the spatial distribution of the fluxes.

P15, L1-2: While I agree that prescribing the stratospheric CH4 fields based on satellite observations might help, this will lead to transport that is not mass conserving, which is a problem for flux inversion. Please comment. Perhaps also mention that MIPAS and ACE-FTS are both good candidates for such an approach, but the former is not flying right now, and the latter has already been flying for 11 years and there is no replacement in sight.

[Figure]

Typographical/language comments:

P3, L9: add "the" before "assimilation"

Table 1: The sign on the latitude of Darwin is wrong in this table.

P4, L11: Add degree symbol on both 4 and 5.

P5, L5: "data WERE available" (plural)

P5, L13: "and initial conditions" -> "and used as initial conditions"

There is no reference to Appendix A2 in the text.

P5, L18: "test the dependence of our results ON the"

p6, L1, L5, and a few other places: "emissions seasonality" isn't quite right. It should either be "the emissions' sensitivity" or "the seasonality of the emissions".

p11, L6: emissions -> emission

---

## Author Comment (AC1) · 18 Aug 2016

We thank Referee #2 for their comments.

The primary change in the updated manuscript is a reprocessing of the TCCON tropospheric methane ($CH_4$) column-averaged dry-air mole fractions (DMFs), which is described in detail in added supplement, "Updates to Tropospheric Methane Data" (Appendix A). Although some of the regression statistics and comparisons have changed as a result of measurement updates, the main conclusions, the mismatch in tropospheric seasonality and the dependence of the stratospheric contribution error on

tropopause height, remain the same.

In our responses below, page and line numbers included refer to the previous discussion draft. Appendices are referred to based on their order in the revised manuscript, and their headings are noted to avoid ambiguity.

**It lacks precision in the text in many places (see specific comments), so are legends of some figures. Several important sentences, often when synthetizing results are confusing and not clear and make the reading not fluid at all with this version (see specific comments). I find the result section, a bit too descriptive, not providing systematically explanations or hypotheses for the inferred results. This has to be improved as it is not done either in the discussion part.**

When discussing values presented in figures, the text now repeats these values more consistently. We have characterized the results more systematically, with greater detail and hypothesized explanations given for each feature. In addition to changing the wording where requested in the specific comments, we have altered ambiguous phrases, removed redundancies, and partitioned long sentences to make explanations simpler and more straightforward. We have also described and removed inconsistencies in terminology for greater clarity. The discussion of the figures in Section 3 has been updated to delineate the results quantitatively and with more detail. We also have made existing explanations more evident and provide additional hypotheses for results.

**About the hypotheses, for instance among several other things reported below, I wonder why the aseasonal run disable the seasonal emissions and scale up the rest instead of prescribing the annual mean of seasonally changing sources ? This is strange as it changes the spatial distribution of emissions on the top of the suppression of seasonality.**

We agree that producing aseasonal emissions by changing the seasonally varying

fluxes to be constant throughout each year for each grid box would be ideal. Unfortunately, the model infrastructure made such a simulation difficult to execute as it required the emissions code to be re-written, risking differences due to compiling changes. Thus the scaling technique was developed as an alternative to assess first-order impacts of emissions seasonality. We have added this explicitly as a limitation that should be improved on in the future. However, most of the notable results, especially the phase lag in the tropospheric seasonality, are consistent between the model runs despite any differences in the spatial distribution of emissions. This demonstrates the robustness of our conclusions regardless of the emissions fields used. Additionally, the analyses comparing the base and aseasonal simulations are aggregated on zonal or hemispheric scales and therefore should not vary because of the spatial differences of their emissions at smaller scales.

**Abstract : "large number of highly variable sources" not all methane source are highly variable. On what scale ? And sinks ? I suggest because of a large number of uncerain sources and sinks.**

The phrase, "highly variable sources" has been removed for conciseness.

**Page 2 : lines 1-5 : the words "atmospheric inversion" should appear somewhere in this paragraph.**

The term "atmospheric inversion" has been added for clarity.

**Lines 16-17:Do they have the same bias as aircraft observations of clear-sky only measurements (aircraft do not fly in bad weather conditions)? It is worth noticing this issue somewhere.**

TCCON FTS instruments do not make measurements in rainy or completely overcast

weather, which is now noted.

**Lines18-20 : Fraser et al : how did they do that ? did they account for observation systematic errors as well ? Please be more precise when quoting papers. Idem for Wecht et al.**

Additional descriptions of the approaches of Fraser et al. (2013) and Wecht et al. (2014) are now included in the introduction, and greater detail was added for several other references elsewhere in the manuscript. While Fraser et al. (2013) performed a variety of observing system simulation experiments (OSSEs) to test measurement and sampling biases, their focus was the information content of different types of observations in relation to atmospheric inversions. We have included their sector and regional error reduction results for the reader's reference.

**Lines 33-35 : ambiguous sentence. Please rephrase. Indeed tropospheric CTM do not reproduce well stratospheric transport...**

"Insofar as," has been changed to, "Provided that," to make the conditional aspect of the sentence more clear and reduce ambiguity.

**Page 3 : Line 6 : "systematic model biases" : strange expression. Maybe systematic errors would be enough. What about the random part or errors? Do you address this as well ? Please reformulate.**

By biases, we refer to the measurement-model mismatch due to inaccuracies inherent in the model; we agree that "systematic errors" also relays this meaning and have changed the wording. Because the focus of this work is on systematic differences between observations and the model, we do not quantify random model error except to note how the scatter and goodness of fit of the linear regression analyses compare

between subsets of data (e.g. Northern vs. Southern Hemisphere and $X_{CH_4}^t$ vs. $X_{CH_4}$).

**Line 6 : "seasonal cycle and spatial distribution of CH4" concentrations ? emoissions ? please be more precise.**

This phrase now reads, "the seasonal cycle and spatial distribution of $CH_4$ DMFs" for clarity.

**Line 15 : it would be good to briefly recall how the TCCON total columns are inferred. In particular, what is the influence of the modelled CH4 profile used in the retrieval (as a prior) on the final product. As this profile comes from a model, it would be worth commenting on this considering the topic of the paper.**

A brief description of the TCCON total column retrievals is now included at the end of the first paragraph of Section 2.1. In addition, a detailed description and references for the $CH_4$ a priori profiles have been added to the text. In testing the influence of the TCCON prior profiles in their comparisons to aircraft in situ profiles, Wunch et al. (2010) found that the total column retrievals using TCCON a priori profiles produced the same calibration values as those using the aircraft profiles as priors.

The newly added Appendix A, "Updates to Tropospheric Methane Data," includes a more detailed description of how the $X_{CH_4}^t$ measurements are determined, processed to address spectroscopy-related errors, and calibrated to in situ aircraft profiles. The consideration of the chosen TCCON priors on the model comparison is addressed by smoothing the GEOS-Chem profiles using the TCCON scaled priors, as described in Appendix B3, "Model Smoothing for Measurement Comparisons." The strong agreement between the integrated and smoothed GEOS-Chem column-averaged $CH_4$ DMFs also supports a negligible influence of the TCCON priors the results (Fig. 12).

**L16 : "precise" : please be more quantitative here or remove the word. How precise compared to surface networks for instance? how is your data uncertainty estimated ?**

In addition to the details provided in response to the previous comment, Wunch et al. (2015), which describes in detail the determination of the TCCON total column uncertainty budgets and quantitative measures thereof, has been added to the references cited on p.3 l.16. A sensitivity study to assess uncertainties related to a priori profiles, spectroscopy, and instrumentation found aggregated $X_{CH_4}$ errors to be below 0.5%, or about 5 ppb (Wunch et al., 2015). Appendix A, "Updates to Tropospheric Methane Data," provides more details on the tropospheric measurement uncertainties, including $X_{CH_4}^t$ precision values and the aircraft in situ calibration curve, for reference.

**Page 4 : Lines 14-15 : please provide a reference for emissions and OH. Do they vary inter-annually ? For OH concentrations, what is your ratio NH/SH ? More precisions are needed here. Indeed you release emissions evey hour but their time evolution is monthly or annually probably. Please precise this not to le the reader think that we know methane emissions with an hourly time step !**

References were cited for the "default" offline $CH_4$ simulation, which included a description of these fluxes. We have since added details and references for each of the emissions categories have been added for the reader's convenience. The list of emissions, which were grouped by time evolution (annual, monthly, and daily), now includes references and additional details that should make the time scales of their variability more apparent to the reader.

The Northern to Southern Hemisphere ratio of 1.0 (monthly range of $0.975 - 1.02$, applying a six month lag in the Southern Hemisphere) is consistent with the ratio of $0.97 \pm 0.12$ found by Patra et al. (2014). The tropospheric OH are monthly-averaged output from a GEOS-Chem tropospheric chemistry simulation (Park et al., 2004). The

description of tropospheric OH and stratospheric loss parameterization fields now include references.

**Section 2.2 : It would be useful to position GEOS-CHEM with other transport models based on previous Transcom-like experiment (e.g. : Patra et al., 2011): is it a "fast" model ( inter-hemispheric exchange time ?), what about stratosphere/troposphere exchange time ? ... It would be very useful for other modellers to use the results of the paper.**

Unfortunately, Patra et al. (2011) does not disaggregate the quantitative metrics asked for by the reviewer by model in the TransCom-$CH_4$ model comparison. Based on Fig. 8 therein, the interhemispheric exchange time in GEOS-Chem appears near the model median and slightly below observations over the 1996-2007 time series, which we have added to the conclusions for the reader's reference.

**Page 5 : Line 1-2 : this first sentence needs precision : what is GGG2014 ? What is GEOS5 ? Acronyms have to be defined and explained**

GGG is the name of the software and not an acronym. GGG2014, the current version of the TCCON retrieval software package, is described more fully in Section 2.1, where it is first introduced, to avoid confusion. The full name for the GEOS-Chem GEOS5 meteorology is now included on p.5 l.2.

**Lines 10-15: the choice to disable the seasonal emissions and scale up the rest is strange as it changes the spatial distribution of emissions on the top of the suppression of seasonality. Why not prescribing the annual mean of seasonally changing sources ?**

Please see the above response to the related general comment.

**Line 15 . What is "TCCON daily median scaled priors" ? you need to provide more details here.**

GEOS-Chem smoothed column-averaged DMFs were only calculated for days in which TCCON measurements were made and were smoothed using solar zenith angles, vertical scaling factors, and surface pressures for TCCON measurements used in the comparisons. The added discussion of the TCCON retrieval in Section 2.1 provides a description of the vertical scaling factor that clarifies subsequent references. To further lessen confusion, this sentence has been changed to, "For comparisons with column measurements, model vertical profiles were smoothed with corresponding TCCON $CH_4$ averaging kernels, interpolated for the daily mean solar zenith angles, and prior profiles, scaled with daily median vertical scaling factors and interpolated to the daily mean surface pressures measured at each site, following the methodology in Rodgers and Connor (2003) and Wunch et al. (2010)."

**What is the influence of these "priors" on the TCCON products and on the comparison proposed here.**

As mentioned in the note above referring to the comment about p.3 l.15, Wunch et al. (2015) describes sensitivity experiments to assess the systematic errors in the TCCON retrievals that could potentially result from the a priori profiles. As Fig. 10 of that document illustrates, shifting the trace gas profiles down 1 km in altitude and increasing the temperature by 1 K and pressure by 1 hPa throughout the vertical profile each alter $X_{CH_4}$ by about $0.05 - 0.1\%$. For the purposes of this work, the strong agreement between the GEOS-Chem column-averaged $CH_4$ DMFs and those smoothed using the TCCON scaled a priori profiles, as described in Appendix B3, "Model Smoothing for Measurement Comparisons," demonstrates the unlikelihood of the TCCON priors being the reason for the measurement-model disagreement (Fig. 12).

**Line 20 : "While XtCH4 20 changed slightly" : how much is the change ? please provide % for instance. Why only testing above levels ? please provide explanations.**

The median change in $X_{CH_4}^t$ of about 1 and 5 ppb for a respective one and two-level increase in tropopause is now stated.

Accurately representing GEOS-Chem's tropospheric column for the purpose of comparison to measurements depends on setting the tropopause so that the calculation from model output is consistent with the way the model defines the troposphere. Shifting the tropopause level allowed us to test the degree to which calculating $X_{CH_4}^t$ using the daily average tropopause could bias the comparison. Furthermore, because the vertical gradient of $CH_4$ is steepest across the UTLS, choosing a lower tropopause level would change the vertical integration much less than choosing a higher level. Thus, integrating to higher pressure levels would provide a better measure of sensitivity to the integration tropopause height chosen.

**Line 24: "small". Please be more precise. Remain within ±5 ppb for instance ? Idem for larger NH changes : 'varies from -10 to +13 ppb ?**

Quantification of the seasonal cycle has been added: "within $\pm 4$ ppb" for the Southern Hemisphere and "varies between $-10$ and $+13$ ppb" for the Northern Hemisphere troposphere.

**Page 6 : Lines1-2 : what do you mean by " common " ? Why the age of air increases when seasonality is supressed ? Please provide more clear explanations.**

Because the transport of tropospheric air to the stratosphere air is governed by vertical

ascent in the tropics (Brewer, 1949; Dobson, 1956), stratospheric air has a shared source of $CH_4$ that lessens the interhemispheric gradient seen in the troposphere (Boering et al., 1995, 1996). The age of air does not increase with dampened seasonality; rather the signal of tropospheric seasonality in a given parcel of air dissipates as its residence time increases (Mote et al., 1996). We now discuss this in more detail in Appendix A, "Updates to Tropospheric Methane Data."

**Line 3 : "relatively short" : please provide an estimate**

The model's equilibrium lifetime of $CH_4$ in the stratosphere is about 22 months, which we now state in the text.

**Page 7 : Line 9-10 : tropospheric slope does not seem lower than one for southern stations. Indeed it seems there is a little north-south gradient in the tropospheric slopes. Did you investigate it ?**

The tropospheric slope did not have an interhemispheric difference prior to the $X_{CH_4}^t$ update. However, with the updated $X_{CH_4}^t$ observations, the plots show interhemispheric differences in both $X_{CH_4}^t$ and $X_{CH_4}$. These Northern and Southern Hemisphere comparisons between TCCON and GEOS-Chem are described fully in Section 3.

**Figs 4 : this figure is not enough analysed. You do not comment : - the negative bias of GEOS-CHEM at most sites for the trospospheric & total columns (4ab) - the fact that stratospheric columns of GEOS-CHEM seems underestimated for more southern sites and overestimated for more northern sites (4c) - possible reasons for the poorer agreement in the stratosphere.**

The discussion of Fig. 4 now includes a systematic description of the plots, with associated hypotheses. The underestimation of $CH_4$ concentrations in GEOS-Chem has

been documented elsewhere. In the TransCom-CH4 model comparison, GEOS-Chem $CH_4$ concentrations were lower than the model median, and when using the same OH fields much lower than the range of other models (Patra et al., 2011). The negative bias was previously described as an offset when discussing the impact of the aseasonal simulations, and we have added that the direction of the offset (i.e. GEOS-Chem is systematically low) and provide a hypothesis for why the offset changes between simulations.

The stratospheric contribution of $CH_4$ increases from the equator to the poles due to the zonal gradient in tropopause height. We have added a discussion of the zonal gradients in the measurement-model differences in $X^t_{CH_4}$, $X_{CH_4}$, and the stratospheric contribution. We also directly compare the agreement (both slopes and $R^2$ values) across plots and hypothesize why correlations vary for different vertical levels.

**You may also consider two slopes, one for the southern stations (larger thabn 1) and one for the northern stations (smaller than 1) on fig 4a, or a non linear continuous decrease of the slope from south to north. Why only keeping a global slope ?**

We had plotted regression lines across all sites in Fig. 4 and listed in the text the individual hemispheric regression results for the stratospheric contribution. However, we agree that providing regression equations for each hemisphere is more illustrative. Regression lines and equations for Northern and Southern Hemispheres now appear on the plots in Fig. 4.

**Page 8 Line 3-6 : any possible explanation for the differences with ACE ?**

The structure of the differences with ACE-FTS measurements illustrated in Fig. 5 demonstrate that the cause is systematic to the model. GEOS-Chem is too low above the tropical tropopause in both boreal spring and fall and too high in boreal spring directly above the Northern Hemisphere mid-latitude tropopause and in the Southern Hemisphere high altitudes. The ACE-FTS data gaps in the tropical troposphere prevent assessing whether vertical ascent into the stratosphere is too weak. Because the stratospheric loss parameterization is produced from NASA Global Modeling Initiative (GMI) model output, biases in the rate of loss could result from intra-model differences in transport schemes. A more thorough description of Fig. 5 and possible explanations for differences have been added to the paragraph on p.8 l.3.

Additionally, the ACE-FTS climatology plotted in Fig. 5 is an older version of the measurements (v. 2.2, Jones et al., 2012), which also could impact some of individual grid box differences; however, a comparison to the monthly means of the v.3.5 $CH_4$ DMFs (which are used in the $X_{CH_4}^t$ calculation) indicate that the data version likely would not change main features of Fig. 5.

**Line 12-14 : "As the effective..pressure heights" : unclear sentence. Please rephrase.**

The sentence has been rephrased: "The disagreement exhibits a large spread for relatively few tropopause pressure heights because the model's effective tropopause, that is, the pressure level at which the model divides the troposphere from the stratosphere in GEOS-Chem, is defined at discrete grid level pressure boundaries."

**Page 10 line 10 : "production" or emissions ?**

We infer that the referee meant p.10 l.5 and have changed "production" to "emissions." Otherwise, we do not understand the comment in the context of p.10 l.10.

**Page8-9 Line 15-4 : the part about troposphere is confusing as figure 6b shows similar trend for stratosphere and troposphere but you mention in the text much**

**lower sensitivity. Please clarify this section.**

While the slope is similar between the stratospheric contribution and tropospheric column, the correlation coefficient is higher for the stratosphere than the troposphere, meaning that the tropopause height can explain a higher percentage of the variance in the measurement-model mismatch for the stratospheric contribution versus $X^t_{CH_4}$. Moreover, despite the similar slopes, the direction of the relationship with respect to $\Delta CH_4 = 0$ is opposite: Fig. 6 shows that the mismatch increases as the tropopause height decreases for the stratospheric contribution (with the model's contribution of the stratosphere becoming larger than that of the measurements) and vice versa for the tropospheric mismatch (with the measurements and model showing better agreement when the tropopause height is lower). These points of clarification have been added to Section 3.1.

**Page 10 Line 5 : "production" do you mean emissions as there is no methane 3D production in the atmosphere ?**

As stated above, "production" has been changed to "emissions."

**More, your statement brings more the summer large wetland emissions as an explanation for the phase of the modelled signal than the loss which should produce more a fall maximum as in surface observations (although Par falls is not the best example to discuss seasonal variations as the signal is complex). Please clarify.**

We agree that the emissions are likely the main driver of the model's surface seasonality, and we have removed, "and loss," for clarity.

Park Falls was chosen because of the TCCON sites that also have surface observations, the $X^t_{CH_4}$ seasonality most closely matches the Northern Hemisphere mean

shown in Fig. 7; thus Fig. 8 provides a good basis to compare surface and tropospheric column measurements. While the site does have a complicated seasonality near the surface, we find it notable that GEOS-Chem is able to capture several of those features, especially the local minimum in October, but still deviates from the observations, as we note on p.10 l.8.

**Page 11 : Lines 1-3 : please develop a bit why you discard OH as an hypothesis to explain the inferred changes?**

The sensitivity experiments we ran tested a number of different OH (as well as emissions and meteorology) fields, which included scaling the default OH fields and using different scalings of the "Standard Chemistry + Biogenic VOCs" OH output (which is now described in more detail in Appendix B1, "Equilibrium Sensitivity Experiments"). The seasonal phase shift appeared in all simulations, regardless of OH used, although the seasonal cycle amplitude and the shape of the springtime maximum varies between simulations. A table delineating these simulations has now been added to Appendix B1. Additionally, p.11 l.2 now refers to a figure, also in Appendix B1, which illustrates the tropospheric seasonality of each of these simulations, as well as deviations from the base simulation.

**Lines 7-8 : "The model sensitivity kernel implicitly includes.. " well do you mean variance matrices associated with observations ? with prior emissions ? Indeed, transport errors are generally implicitly include in atmospheric inversions by inflating observations errors but are not part formally of the variance matrix of emissions. Lines 7-8 : "which are compounded if vertical levels are subject to different errors" Confusing sentence. What do you mean ? pleas clarify.**

The sensitivity kernel refers to the linear operator that maps $CH_4$ emissions to $CH_4$ concentrations; together with the error covariance matrices, the sensitivity kernel is used

to calculate the gain matrix used in inversions to determine posterior emissions. The literature is inconsistent in how to refer to this operator; thus we use the term "sensitivity kernel" because we thought it describes the function of the operator: to provide the change in the $CH_4$ concentration resulting from a perturbation to emissions for a given grid box. The response of modeled $CH_4$ concentrations to changing emissions depends on the model's transport and chemical loss, as well as assumptions about when and where fluxes occur. Therefore, uncertainties in these terms will be implicitly included in the sensitivity kernel. We have clarified what we refer to as the sensitivity kernel after the introduction of the term and have altered the wording to make the logic more linear. The subsequent sentence now states, "The model's stratospheric response to emissions perturbations differ from that of the troposphere and are subject to different transport and loss errors."

**Page 12 : Line 6 'Although the stratosphere accounts for about 30% ' if you refer to top panel of figure 9, I suggest up to 35 % (JJA)**

The top panel of Fig. 9 is the fraction of total emissions that are seasonally varying (that is, from wetlands, rice paddies, biomass burning events) in GEOS-Chem. The 30% value cited is the mean fraction of the total column of $CH_4$ (in units of molec·cm$^{-2}$) that exists in the stratosphere.

**Fig9 : The legend of figure 9 is unclear. Top panel : fraction of what ? Bottom panel : the orange curve is a difference or the error of the aseasonal ? Unclear.**

The upper panel of Fig. 9 is now labeled. As the caption reads, the orange curve is the difference between base and aseasonal simulation tropospheric columns. The label provides a qualitative description to improve on the originally submitted figure after we received feedback that the label, which explicitly stated that the curve is the tropospheric difference, was unclear.

**line 10 : "The seasonality of the stratospheric error will therefore distort the inversion mechanism and thus posterior emissions estimates." : well only if these error are not included in the inversion variance matrices. I would be more confortable writing may distort or precise the conditions of influence of the seasonality in the stratospheric signals on surface emissions through inversions.**

The uncertainties associated with transport are generally accounted for in inversions as a subjective percent error applied to all grid boxes, which would not capture the stratospheric errors presented here. Incorporating stratospheric uncertainties into the error covariance matrix would require a thorough characterization of those errors as a function of longitude, latitude, altitude, and month. Such efforts would be indispensable in improving the forward model, but our concern is that the error covariance matrix is not equipped to correct for these systematic biases. The conditions of influence of the stratospheric seasonality are delineated in the subsequent text.

**Line 11 : "product of transport errors" : how did you evaluate the possibility of issues related to OH radicals ?**

We infer that this refers to p.14 l.8. As mentioned above, we ran sensitivity experiments testing various OH fields, and these are now described in more detail in Appendix B1, "Equilibrium Sensitivity Experiments." Because the tropospheric phase shift appeared in all simulations, regardless of OH used, we believe that the tropospheric OH cannot account for the error in seasonality. We have added a table describing these simulations and a figure that plots the tropospheric seasonality of each of these simulations to Appendix B1.

**Lines 10-12 : it is never mention except in caption of figure 4 that 't' in CH4t refers to troposphere**

[Figure]

The description of the tropospheric $CH_4$ columns introduces the superscript $t$ notation to indicate a tropospheric column-averaged DMF (p.4, l.1).

**"their emissions are very uncertain" : you may quote a recent estimate such as in Kirschke et al. 2013 or IPCC.**

The 2000-2009 range for natural wetlands given by Kirschke et al. (2013) (142-284 TgC·year$^{-1}$) is now included.

**Page 13 : Lines 7-8 : "both the magnitude and seasonality of the difference is significant" : the unit (tons) makes it difficult to say so. There is obviously a sensitivity if transport error shift the seasonality but what does it give in terms of ppb ? or in terms of % of initial emissions ? This would be more clear for the reader.**

The value plotted in Fig. 10b is a sensitivity, in units of kgCH$_4$ per 1 ppb, and can be thought of as the change in emissions needed to increase the DMF at the surface by 1 ppb. Because the seasonality of wetland emissions is such that many grid boxes have no wetland emissions in the winter (Fig. 10a), the emissions related to the phase lag as a percentage change from the prior would produce infinite or very large percentages. Thus, presenting the values as percentages would provide a large range of values but very little information about the absolute emissions. We have set the units of Fig. 10 a and b equal, to make the comparison more clear to the reader. Additionally, we have updated the calculation as the sensitivity to 1 ppb increase in $CH_4$ over the tropospheric column, not merely at the surface, as the focus on this analysis is the assimilation of column data.

**Lines 8-10 : "The largest disagreements between measured and modeled Xt occur ... than annually." This sentence is unclear to me. Please rephrase.**

This sentence has been expanded and clarified, "Large differences between measured and modeled $X_{CH_4}^t$ are concurrent with low emissions from seasonal sources. The adjustments to prior emissions produced by larger measurement-model disagreement that occur when seasonal sources are a small fraction of total emissions will overestimate posterior emissions from aseasonal sources. Thus these seasonal errors will bias source apportionment toward emissions that do not vary on timescales shorter than annually."

We have also added a more explicit description of the relationship between the seasonality of measurement-model disagreement and that of emissions that vary monthly before the discussion of Fig. 10.

**Page 14 : line16 "the meridional gradient" of what ? emissions ? concentations ?**

This sentence has been changed to clarify that we refer to the meridional gradient of $X_{CH_4}$.

**Unit of figure 12 ? Kg/yr ? Maybe change to Tg/gridbox or Tg/yr/∘latitude ?**

Figure 12 plots CH$_4$ column-averaged DMFs in units of ppb, as described on the labels. If referring to Fig. 12, however, the units on the figure are listed as "$\Delta CH_4 (Tg\,mo^{-1})$." The caption has been changed from "Tg" to "summed over each zonal band, in Tg·mo$^{-1}$" for consistency.

**Conclusions Line 3 : re-precise in the start of conclusion the you used GEOS-CHEM and what are XCH4 and XtCH4 as it has to be readable by itself.**

The phrase "retrieved and modeled $X_{CH_4}$ and $X_{CH_4}^t$" has been changed to "TCCON and GEOS-Chem pressure-weighted total and tropospheric column-averaged DMFs

of $CH_4$, $X_{CH_4}$ and $X_{CH_4}^t$" to be more readable.

**Page 15, lines 1-5 : If stratospheric ch4 is largely independent from tropospheric CH4, is it worth developing full tropospheric and stratospheric chemistry models or prescribing stratospheric CH4 based on satellite observations is enough ?**

The insensitivity of the stratosphere to perturbations in tropospheric $CH_4$ suggest that prescribed stratospheric $CH_4$ could be prescribed in such a way that ensures mass conservation. For example, the stratospheric fields could be scaled according to the mass flux from the troposphere. As models develop their representation of stratosphere-troposphere exchange, however, the conservation of mass will need to be more carefully considered. Thus, more developed linear schemes for stratospheric $CH_4$, such as the UCX mechanism we cite or Slimcat (Monge-Sanz et al., 2013), could provide computationally inexpensive ways to set stratospheric $CH_4$.

**References**

[revised manuscript text omitted]

---

## Author Comment (AC2) · 18 Aug 2016

We thank Referee #3 for their comments.

The primary change in the updated manuscript is a reprocessing of the TCCON tropospheric methane ($CH_4$) column-averaged dry-air mole fractions (DMFs), which is described in detail in added supplement, "Updates to Tropospheric Methane Data" (Appendix A). Although some of the regression statistics and comparisons have changed as a result of measurement updates, the main conclusions, the mismatch in tropospheric seasonality and the dependence of the stratospheric contribution error on

tropopause height, remain the same.

In our responses below, page and line numbers included refer to the previous discussion draft. Appendices are referred to based on their order in the revised manuscript, and their headings are noted to avoid ambiguity.

**My first major concern is that the fluxes used for the "Base" case do not actually match the total column TCCON measurements all that well. This can be seen somewhat by the top row of scatter plots in Figure 4. The correlation between the the total column simulated by GEOS-Chem has a correlation with the TCCON measurements of 0.86, which is even a bit lower than the correlation of the tropospheric columns, which are arguably more relevant for flux inversions. But more worrying, in Figure 7 it can be seen that the seasonality of the total column across the TCCON northern hemisphere sites considered is completely wrong. This inability to capture the seasonal cycle in the total column means that only limited conclusions can be drawn from assessing the (slightly different) mismatch in the two parts of the column. Thus I think the main weakness of this paper is the choice of fluxes used for the forward simulation.**

We chose to use the default emissions provided for the GEOS-Chem offline $CH_4$ simulation to demonstrate how systematic errors in the vertical profile of $CH_4$ (which are caused by parameters that do not vary interannually, namely OH fields and transport schemes) can alias into the optimized emissions resulting from an assimilation of total column measurements into an atmospheric inversion. This analysis is a sensitivity study on how model biases can alias into emissions optimization. Thus, the choice of emissions would not drive results unless those emissions are somehow causing the systematic biases. The aseasonal simulation was set up as an experiment to determine if the seasonality of emissions was causing the tropospheric phase lag observed in the base simulation. As Fig. 7 illustrates, the seasonal phase was consistent between simulations even as the amplitude changed, which demonstrates that the cho-

sen emissions fields do not drive the main result of this analysis.

**These fluxes are only listed in terms of categories, with no itemization of which anthropogenic inventory (I guess EDGAR4.X?), which "other natural emissions", or which model was used for the very important seasonal wetland and rice fluxes. At very least this needs to be amended and clarified.**

References were cited for the "default" offline $CH_4$ simulation, which included a description of these fluxes. We have since added details and references for each of the emissions categories have been added for the reader's convenience.

**It's fine that the fluxes are added to the model at 60 second increments, but I guess that aside from fires and wetlands/rice the fluxes are constant throughout the year? Or did you employ a diurnal or weekly or annual cycle?**

The list of emissions, which were grouped by time evolution (annual, monthly, and daily), now includes additional details that should make the time scales of their variability more apparent to the reader.

**And what about the OH fields? Is there a reference for where these came from? Have they been optimized via methyl chloroform or similar?**

Optimized OH fields were not available for GEOS-Chem, which led to the OH sensitivity experiments to test the dependency of $CH_4$ DMFs on the magnitude, seasonality, and distribution of tropospheric OH. These experiments are described in Appendix B1, "Equilibrium Sensitivity Experiments". The Northern to Southern Hemisphere ratio of 1.0 (monthly range of $0.975 - 1.02$, applying a six month lag in the Southern Hemisphere) is consistent with the ratio of $0.97 \pm 0.12$ found by Patra et al. (2014). The tropospheric OH are monthly-averaged output from a GEOS-Chem tropospheric chemistry

simulation (Park et al., 2004). The description of tropospheric OH and stratospheric loss parameterization fields now include references.

**What would have been a more relevant choice for this type of study would be to use optimized fluxes, resulting from an atmospheric inversion using the same model. There are a few groups working on methane inversions with GEOS-Chem, so such fluxes should not have been difficult to find through collaboration. Then you would have been able to start with a seasonal cycle in the column that is actually consistent at the TCCON sites, assuming that the TCCON sites were assimilated in the inversion. This would have made the analysis more relevant, and it would be my strongest recommendation for improving this study.**

As you note, most of the recent optimized emissions that result from atmospheric inversions, especially those using GEOS-Chem as the forward model, assimilate TCCON total column measurements. Using these fluxes would make the measurement to model comparisons, and thus their correlations, no longer independent, and the statistics would be less meaningful.

Moreover, using optimized fluxes may not improve the seasonality of the mismatch. Fraser et al. (2011) compared TCCON total columns to GEOS-Chem run with posterior fluxes, which were derived from an inversion using GOSAT total columns and surface measurements, and found a seasonally-varying measurement-model mismatch that fell between $\pm 20$ ppb (Fig. 6 of that paper). We agree that work that compares optimized fluxes from atmospheric inversions that assimilate data at various vertical levels would be very informative, and this approach would be an important next step.

**Another concern related to the choice of fluxes relates to the method used for the aseasonal simulation. The manuscript describes that the seasonal fluxes (fires, wetlands, and rice) were "disabled" (I assume this means set to zero?), and then**

**the other fluxes were scaled up to maintain the fluxes and the approximate (but certainly not exact, as showns in Figure 11) latitudinal distribution. Why not simply use an annual mean of the variable fluxes? Then you are not changing two things at once (geographic distribution and temporal variability) and attempting to attribute the changes to only one of the factors.**

We agree that producing aseasonal emissions by changing the seasonally varying fluxes to be constant throughout each year for each grid box would be ideal. Unfortunately, the model infrastructure made such a simulation difficult to execute as it required the emissions code to be re-written, risking differences due to compiling changes. Thus the scaling technique was developed as an alternative to assess first-order impacts of emissions seasonality. We have added this explicitly as a limitation that should be improved on in the future. However, most of the notable results, especially the phase lag in the tropospheric seasonality, are consistent between the model runs despite any differences in the spatial distribution of emissions. This demonstrates the robustness of our conclusions regardless of the emissions fields used. Additionally, the analyses comparing the base and aseasonal simulations are aggregated on zonal or hemispheric scales and therefore should not vary because of the spatial differences of their emissions at smaller scales.

**My next major concern is related to the numerics of how the stratospheric and tropospheric model columns are divided. I do not understand how the the statospheric column-integrated dry air mole fractions have values around 30-100 ppb (from Figure 4). This seems very, very low. Looking at the prior profiles from Wunch et al. (2011), Figure 2, the stratospheric values of CH4 range from 500-1800 ppm. I am not sure if this can be explained by the weighting with the pressure-weighted averaging kernel, as the methane column averaging kernel is actually rather flat (from Figure 4, Wunch et al., 2011). Also from Figure 1 of Saad et al. (2014), the only mixing ratios of stratospheric methane less than**

**even 500 ppb seem to be over 40 km or so, which is far above even the highest tropopause. I had postulated that perhaps you had calculated the mixing ratio not in parts per million molecules of stratospheric air but rather of total column air (in which case it should have been explained). Although I would not advocate for such an approach, in that case the stratospheric partial column dry air mole fraction could be added directly to the tropospheric dry air mole fraction to get the total column dry air mole fraction. Looking again at Figure 4, this is clearly not the case: the tropospheric column is clearly larger than the total column. This needs to be clarified.**

You correctly postulated that the stratospheric contribution is calculated in reference to the total column of air. This was done for both practical and conceptual reasons. TCCON $X_{CH_4}$ and $X^t_{CH_4}$ are processed to remove various spectroscopic biases and calibrated to in situ aircraft profiles, now described in Appendix A, "Updates to Tropospheric Methane Data." Thus, using these column-averaged DMFs instead of the $CH_4$ columns in our proxy for stratospheric air ensures measurement biases are not the cause of any measurement-model mismatch.

Conceptually, because this paper focuses on how the model's stratospheric contribution to the total column can alter the conclusions made about tropospheric trends, we determined that stratospheric $CH_4$ over the total column of air would be more relevant than the stratospheric partial column of $CH_4$. We agree that if the purpose of this work was to assess modeled stratospheric profiles, the stratospheric partial column would be more appropriate. Because the stratosphere has less $CH_4$, the stratospheric contribution depresses the total column value, so the tropospheric column average should be larger. We frame the stratospheric contribution as positive number to make the value more intuitive: a larger stratospheric contribution indicates the influence of the stratosphere on the total column is greater. The stratospheric contribution is also represented as a positive number for visual clarity; applying a sign change to the stratospheric contribution in Fig. 4 and adding it to $X^t_{CH_4}$ does reproduce $X_{CH_4}$.

We have updated the wording of the definition of the stratospheric contribution on p.5 l.21 to remove the ambiguity of how the stratospheric contribution is calculated. Additionally, we have added an appendix with the derivation of the stratospheric contribution, "Derivation of Stratospheric Contribution" (Appendix C). We have also changed usage of "stratosphere" to "stratospheric contribution" throughout the text for contexts in which the ambiguity could be confusing.

**One other concern was the consistency of the model tropopause with that from the TCCON retrievals. You mention testing the impact of moving the tropopause model layer up one or two levels, but this does not allow for potential seasonal or regional variability in the match between the two. At very least the (latitude- and seasonal-dependent) correlation between the model and retrieval tropopause heights should be presented in some way.**

Accurately representing GEOS-Chem's tropospheric column for the purpose of comparison to measurements depends on setting the tropopause so that the calculation from model output is consistent with the way the model defines the troposphere. Shifting the tropopause level allowed us to test the degree to which calculating $X^t_{CH_4}$ using the daily average tropopause could bias the comparison. Because the vertical gradient of $CH_4$ is steepest across the UTLS, choosing a lower tropopause level would change the vertical integration much less than choosing a higher level. Thus, integrating to higher pressure levels would provide a better measure of sensitivity to the integration tropopause height chosen.

Additionally, GEOS-Chem sets the top of the troposphere one level below the vertical pressure level below the tropopause, which we thought could also introduce a bias. We ran a simulation setting the top of the troposphere at the level in which the tropopause exists (now listed in Appendix B1, "Equilibrium Sensitivity Experiments"), essentially shifting the tropopause up two levels, to determine if the choice of the definition of the tropopause changed the distribution of $CH_4$ concentrations. This change did not

improve measurement-model agreement and, as the newly added figure demonstrates, had almost no impact on the seasonality of $X_{CH_4}^t$ (Fig. 15 in the updated manuscript).

We consider other inconsistencies in the model tropopause, such as seasonal or zonal variability, as one of the model errors that can alias into $X_{CH_4}$ comparisons. The calibration of TCCON measurements to in situ aircraft profiles (Wunch et al., 2015) limits any bias that errors in the TCCON tropopause heights could induce in the comparisons with the model. Moreover, the difference between calibrated TCCON and integrated aircraft $X_{CH_4}$ and $X_{CH_4}^t$ values have no correlation to the tropopause heights used to generate the TCCON priors or computed from the aircraft temperature and pressure profiles (uncertainty-weighted $R^2 = 0$).

**P3, second paragraph: This sounds like you're describing atmospheric inversion while going out of your way not to call it "inversion". Or are you referring to optimization only by processed-based scaling of set spatial fields? Please clarify, and if you mean inversion, please say so.**

We infer that the referee meant p.2 second paragraph and have added the term "atmospheric inversion" for clarity.

**P2, L10: The reference to Stephens et al. (2007) here seems not to fit so well - this study was looking at aircraft profiles rather than column-integrated information.**

The reference of Stephens et al. (2007) was included to illustrate the importance of assimilating observations that provide information about the vertical profile to accurately constrain chemical transport models. We agree that p.2 l.10 is not the appropriate location for this point and have moved the reference to the paragraph on p.2 l.22.

**P4, L14: Although I mentioned it already above, there needs to be some citations**

**to describe the model and fluxes used.**

Citations and additional details for the GEOS-Chem offline $CH_4$ emissions, tropospheric OH, and stratospheric loss fields have been added.

**P5, L9-10: In Appendix A1 I coudln't find any real description of the OH sensitivity runs. Do your OH fields have seasonality? This experiment is insufficiently described.**

We have added the following description of the "Updated OH" simulation (Table 2, Fig. 3) to Appendix B1, "Equilibrium Sensitivity Experiments":

> "The updated OH simulation used OH output from a 2012 GEOS-Chem standard chemistry simulation with extensive updates to the photochemical oxidation mechanisms of biogenic volatile organic compounds (VOCs), described in Bates et al. (2016) and references therein. These were converted to 3D monthly mean OH concentrations to conform to the infrastructure of the GEOS-Chem offline $CH_4$ tropospheric loss mechanism. The OH was then scaled by 90% to keep the lifetime above 8 years, and emissions were scaled by 112% to maintain the same balance between sources and sinks in the base simulation. Figure 14 provides zonal averages of the difference between the base and updated OH columns."

We also ran several sensitivity experiments on different OH fields, which included scaling the default OH fields and using different scalings of the "Standard Chemistry + Biogenic VOCs" OH output. A table delineating these simulations has now been added to Appendix B1.

**P5, L15: I was a bit confused here: are the means and medians for all values over the day, over just over those where TCCON measurements were made?**

GEOS-Chem smoothed column-averaged DMFs were only calculated for days in which TCCON measurements were made and were smoothed using solar zenith angles, vertical scaling factors, and surface pressures for TCCON measurements used in the comparisons. The added discussion of the TCCON retrieval in Section 2.1 provides a description of the vertical scaling factor that clarifies subsequent references. To further lessen confusion, this sentence has been changed to, "For comparisons with column measurements, model vertical profiles were smoothed with corresponding TCCON $CH_4$ averaging kernels, interpolated for the daily mean solar zenith angles, and prior profiles, scaled with daily median vertical scaling factors and interpolated to the daily mean surface pressures measured at each site, following the methodology in Rodgers and Connor (2003) and Wunch et al. (2010)."

**In general I found the use of "DMF" to mean "column-integrated dry air mole fraction" to be rather confusing. Flask measurements also measure dry air mole fraction, so DMF on its own does not tell the reader that an integrated column is being discussed. This is found throughout the manuscript and should be clarified.**

The modifier "column-average" now precedes "DMF" unless referring to a surface or profile measurement to maintain consistency and avoid ambiguity.

**Figure 3: The caption says that the stratosphere shows a seasonal cycle of 15 ppb at Park Falls, but in the figure looks like more like 30 ppb. Please explain. I was also surprised to see that Park Falls appears to have a larger seasonal cycle in hte stratosphere than in the troposphere for the Base case. This doesn't make sense to me. Please explain.**

The text cites a seasonal amplitude of 15 ppb, referring to the peak amplitude of the seasonal cycle (i.e. the difference between the peak and the mean). The peak-to-

trough amplitude, which is twice the peak amplitude, would indeed be 30 ppb. We have changed the word "amplitude" to "range" and updated the values accordingly to reduce confusion.

The model's larger seasonal cycle of the stratospheric contribution compared to that of the tropospheric column does not agree with the measurements, as illustrated by Fig. 7. The stratospheric contribution is a function of the gradient across the tropopause and $CH_4$ loss in the stratosphere (Appendix C, "Derivation of Stratospheric Contribution"); thus, model errors in prescribed tropopause height, stratospheric chemistry, and stratospheric transport will impact the seasonal cycle of the stratospheric contribution.

**P7, L13-14: What about the significant figures on the slopes (e.g. 1.1±0.020).**

The extra significant figures on the slope errors were unintended and have been removed.

**P8, L5-7: I'm not sure that Figure 4 shows a good agreement between the stratospheric columns of TCCON and GEOS-Chem. Yes, the clump of points is closer to the 1:1 line, but it hardly forms a line at all. Is the correlation coefficient for this one station really notably higher?**

The wording indicating good agreement has been changed to, "fall most closely to the one-to-one line." The spread across the one-to-one line seen at Lauder is partly due to seasonal variability, as the stratospheric loss parameterization in the model is monthly. Averaging GEOS-Chem daily values to correspond to the ACE-FTS and GEOS-Chem climatologies would make the relationship more compact.

**Figure 5: Again I'm confused about the calculation of the stratopsheric column. For instance, we can see from Figure 4 that the stratospheric column simulated**

[Figure]

**by GEOS-Chem is around 50 ppb. Then looking at Figure 5, ACE-FTS minus GEOS-Chem seems to show a difference of approximately -50 ppb around 45 degrees south. Does this mean that the ACE-FTS measurements are showing close to zero methane? In general there seems to be better agreement between TCCON and GEOS-Chem (Figure 4) than ACE-FTS and GEOS-Chem (Figure 5), but it is difficult to tell from the figures presented. Could you comment on this? How do ACE-FTS and TCCON agree?**

Figure 5 illustrates differences between the $CH_4$ profiles given by ACE-FTS and GEOS-Chem climatologies, not pressure-weighted column averages as in the TCCON comparison. As a point of reference, they correspond to the prior profiles from Wunch et al. (2011) you mentioned in previous comments. Thus, the $\pm 150$ ppb range appertains to the difference of the mean $CH_4$ mole fractions at each pressure level. The ACE-FTS climatology used in Fig. 5 is an older version of the measurements (v. 2.2, Jones et al., 2012), which could impact some of individual grid box differences. However, a comparison to the v.3.5 (which are used in the $X_{CH_4}^t$ calculation) monthly mean $CH_4$ DMFs indicate that the data version likely would not change main features illustrated in Fig. 5.

Because the comparisons between TCCON and GEOS-Chem are for pressure-weighted column averages, the agreement is therefore not directly comparable to ACE-FTS mole fraction differences at individual pressure levels. Agreement between TCCON and ACE-FTS is difficult to quantify because ACE-FTS retrievals provide vertical information solely in the upper atmosphere, and TCCON retrievals provide column averages that, due to the pressure weighting, are dominated by the troposphere. However, ACE-FTS is one of the various platforms used in the development of the empirical model that generates TCCON priors (Wunch et al., 2015), and the stratospheric $CH_4$ profiles it measures are used in the calculation of the TCCON tropospheric $CH_4$ product (Saad et al., 2014).

**Figure 7: I am very surprised to see that the aseasonal simulations have higher**

**seasonal cycles in both the stratosphere and the stratosphere than the base case. Are you sure of this result? What role does the (potential) seasonality of the OH sink have here?**

The larger seasonal amplitude of the aseasonal $X^t_{CH_4}$ is indeed a notable result. The greatest differences, from August through October, result from dampening the large summer wetland fluxes that balance high summer OH concentrations in the base simulation. The larger variance across sites that we note is also indicative that the seasonal amplitude does not increase as drastically at the sub-tropical sites. (We did not include the figure with all site seasonalities because it was visually chaotic, given the many Northern Hemisphere sites.)

The second largest difference, during the spring, could also be a result of the source/sink balance: the aseasonal simulation introduces fluxes in the winter, when the OH concentrations are lowest. As we mention on p.10 l.18, the model may also have an error in phase with the seasonal emissions that produces the reasonable seasonal cycle amplitude in the base simulation troposphere (Fig. 7). We have added to that paragraph a discussion of the interaction between emissions and OH loss.

While the seasonal amplitude of the mean Northern Hemisphere stratospheric contribution is larger for the aseasonal versus base simulation, the maximum difference of their means is only about 2 ppb, which is within the $1\sigma$ standard deviations across sites. This similarity further demonstrates the insensitivity of the model's stratosphere to chosen emissions.

**And what about the sampling throughout the year? Are there enough measurements at Bremen in December and January, or is part of this seasonality a question of shifting sampling throughout the year? Related to this: I assume you are only considering days on which there are TCCON measurements in the model analysis?**

The seasonality of GEOS-Chem is computed from the smoothed pressure-weighted column-averaged DMFs, which incorporate the TCCON scaled prior profiles (see Appendix B3, "Model Smoothing for Measurement Comparisons") and thus require us to consider days on which TCCON measurements exist. While the number of measurements per month is variable throughout the year, all high latitude sites have a time series long enough to extract detrended monthly mean information. Moroever, the sites that are most susceptible to low winter sampling are the five in Europe, which are located in adjacent GEOS-Chem grid boxes. Because we average the seasonality across the Northern Hemisphere, the aggregate of these high-latitude sites would remove any impact that fewer winter measurements have. Figure 1 plots the Northern Hemispheric seasonality without the sites north of $50°$N, Bialystok and Bremen, for comparison. The only sites that are not included in the Northern Hemisphere seasonality are those which began taking measurements less than a year before the end of the model run: Saga and Réunion Island. We have rectified this omission in the text.

**Another surprise here is that that seasonal cycle of the tropospheric and stratospheric columns in the aseasonal case are essentially in phase, yet when the total column is considered, a bimodal seasonal cycle is found. How can this be?**

The stratospheric contribution is the amount by which the stratosphere decreases the total column average (via stratospheric loss and transport). Thus, the stratospheric contribution has an inverse effect on $X_{\mathrm{CH_4}}$ relative to $X_{\mathrm{CH_4}}^t$, and the balance between the stratospheric contribution and $X_{\mathrm{CH_4}}^t$ causes the seasonality in $X_{\mathrm{CH_4}}$. We define the stratospheric contribution more explicitly and include its derivation in Appendix C to prevent confusion.

**P10, L11: I disagree with this statement: it seems that the seasonal cycle of the modelled stratospheric columns precede the seasonal cycle of TCCON by a good month.**

The comparison of the stratospheric seasonality is difficult to assess by eye, but the stratospheric contributions of TCCON and GEOS-Chem are in phase, which is illustrated by the shared inflections point in June and December.

**Figure 8: The smoothing carried out here is not informative. Why not a box and whiskers plot to show how variable the data really are? Also, Park Falls is rather a tricky station with quite a lot of local influence and not a clear seasonal cycle. Perhaps another station would be more informative?**

Park Falls was chosen because of the TCCON sites that also have surface observations, the $X_{CH_4}^t$ seasonality most closely matches the Northern Hemisphere mean shown in Fig. 7; thus Fig. 8 provides a good basis to compare surface and tropospheric column measurements. While the site does have a complicated seasonality near the surface, we find it notable that GEOS-Chem is able to capture several of those features, especially the local minimum in October, but still deviates from the observations, as we note on p.10 l.8. The box and whisker plots with superimposed observations and model data were difficult to follow visually. Instead, to show the variability, we have added to Fig. 8 lower and upper bounds denoting the 25th and 75th percentiles, respectively, of detrended data for each month.

**Also, is the temporal sampling of the model consistent with that of the rather sparse flasks?**

We had compared more frequent "Programmable Flask Package" (PFP) measurements, which have been measured at Park Falls since 2006, and found only slight differences in the seasonal cycle. Because we could not find equivalent in situ NOAA measurements, which we chose because they are on the same calibration scale as TCCON (Wunch et al., 2010), in the Southern Hemisphere, we only plot the flask measurements. Figure 2 plots Fig. 8 with the higher resolution flask data included for your

reference.

**In a broader sense I'm not sure what the real message here is. We see already in Figure 7 that the GEOS-Chem run does a very poor job of representing the seasonal cycle in the NH column: would you expect it to be better at the surface?**

We included surface measurements (a) to demonstrate that the seasonality that we see is not due to some unknown bias in the $X_{CH_4}^t$ measurements and (b) to test whether the phase shift could be due to vertical transport, which would create a smaller lag at the surface, or horizontal transport, which is our hypothesis.

**Figure 9: Please label the plots (especially upper panel).**

The upper panel of Fig. 9 is now labeled.

**Figure 10: The y-axis should have the same scale for the top and bottom figures, even if only part of the range is shown.**

The y-axes of the two subfigures in Fig. 10 have been scaled so that the latitude grid boxes are equal.

**I was also not quite sure about the units here. $10^6$ kg is 0.001 Tg, so the bright yellow (10 $10^6$ kg CH4) is 0.01 Tg CH4. But then in Figure 11 the increments between the seasonal and aseasonal run seem to be rather on the order of 1 Tg CH$_4$ mo$^{-1}$, which is two orders of magnitude higher. Or have I missed something here?**

Figure 10 shows the zonally averaged wetland emissions, while Fig. 11 displays the total difference in emissions. The units in Fig. 10a have been changed to Gg, and the description of units in the caption of Fig. 11 has been changed from "Tg" to "summed

over each zonal band, in Tg·mo$^{-1}$" to prevent confusion.

**P13, L5-6: I did not quite understand the description of what you did here. You write "derived by calculating the total emissions resulting from an increase of 1 ppb of CH4 in each surface grid box". Do you mean by calculating the emissions required to cause a 1 ppb increase in each surface grid box? How often were you adding this increment? Monthly? Do you consider the effect that these emissions have on the concentrations of neibouring grid boxes? Is there a reference that explains this procedure in a bit more detail? Based on what is written here, I could not reproduce the experiment.**

To show the true change in posterior emissions associated with a phase lag, the gain matrix would need to be derived for all grid boxes in the model. Because we did not have the actual sensitivity of $CH_4$ to wetland concentrations, which varies spatially depending on proximity to sources, we estimated that sensitivity as the mass of $CH_4$ associated with a 1 ppb increase in $CH_4$ in the surface grid box. The change in posterior emissions was then calculated as the product of this sensitivity and the fraction of the monthly mean emissions from wetlands in each surface grid box. Figure 10b mapped the difference between this change in posterior wetland emissions and the value in the same grid box three months prior, summed for each zonal band. Because this approach does not include any of the information about transport (as would exist in the linear operator that transforms model emissions to concentrations), we are not able to consider neighboring grid boxes. We have since updated the calculation as the sensitivity to 1 ppb increase in $CH_4$ over the tropospheric column, as the focus on this analysis is the assimilation of column data.

**P14, L8: I don't think you have convincingly shown that the seasonal lag is a function of transport, and not, say, your sink, or the spatial distribution of the fluxes.**

The sensitivity experiments we ran tested the model's response to a number of different emissions, OH, and meteorology fields. The seasonal phase shift in the tropospheric column appeared in all simulations, although the seasonal cycle amplitude and the shape of the springtime maximum varies. We have added a table describing these simulations and a figure that plots the tropospheric seasonality, as well as deviations from the base simulation, of each of these simulations to Appendix B1, "Equilibrium Sensitivity Experiments." We have also removed the sentence in the conclusion referred to by this comment.

**P15, L1-2: While I agree that prescribing the stratospheric CH4 fields based on satellite observations might help, this will lead to transport that is not mass conserving, which is a problem for flux inversion. Please comment.**

The insensitivity of the stratosphere to perturbations in tropospheric $CH_4$ suggest that prescribed stratospheric $CH_4$ would not need complicated adjustment to enforce mass conservation. We agree that the mechanism by which a model would set these $CH_4$ fields in the stratosphere would require careful consideration of how best to ensure the conservation of mass. For example, the stratospheric fields could be scaled according to the mass flux from the troposphere. As models develop their representation of stratosphere-troposphere exchange, however, the conservation of mass will become a more complicated problem. In addition to the UCX mechanism we suggested, a variety of linear schemes for stratospheric $CH_4$ have been tested for other models, such as Slimcat (Monge-Sanz et al., 2013).

**Perhaps also mention that MIPAS and ACE-FTS are both good candidates for such an approach, but the former is not flying right now, and the latter has already been flying for 11 years and there is no replacement in sight.**

This sentence now reads, "satellite observations from ACE-FTS, MIPAS (von Clarmann

et al., 2009), or a compilation of remote sensing instruments (Buchwitz et al., 2015)."
While stratospheric $CH_4$ fields for specific years would be ideal, even a monthly clima-
tology with a secular increase applied would be an improvement on the current loss
parameterization, which are monthly fields that do not vary interannually.

**Typographical/language comments:**
**P3, L9: add "the" before "assimilation"**
**Table 1: The sign on the latitude of Darwin is wrong in this table.**
**P4, L11: Add degree symbol on both 4 and 5.**
**P5, L5: "data WERE available" (plural)**
**P5, L13: "and initial conditions" -> "and used as initial conditions"**
**There is no reference to Appendix A2 in the text.**
**P5, L18: "test the dependence of our results ON the"**
**p6, L1, L5, and a few other places: "emissions seasonality" isn't quite right.
It should either be "the emissions' sensitivity" or "the seasonality of the emis-
sions".**
**p11, L6: emissions -> emission**

The above changes were made, and Appendix A2 (now B2) is now referenced in Sec-
tion 2.2.1

**References**

Bates, K. H., Nguyen, T. B., Teng, A. P., Crounse, J. D., Kjaergaard, H. G., Stoltz, B. M.,
   Seinfeld, J. H., and Wennberg, P. O.: Production and Fate of C4 Dihydroxycarbonyl Com-
   pounds from Isoprene Oxidation, The Journal of Physical Chemistry A, 120, 106–117, doi:
   10.1021/acs.jpca.5b10335, http://dx.doi.org/10.1021/acs.jpca.5b10335, 2016.
Buchwitz, M., Reuter, M., Schneising, O., Boesch, H., Guerlet, S., Dils, B., Aben, I., Armante,
   R., Bergamaschi, P., Blumenstock, T., Bovensmann, H., Brunner, D., Buchmann, B., Burrows, J. P., Butz, A., Chédin, A., Chevallier, F., Crevoisier, C. D., Deutscher, N. M., Franken-
berg, C., Hase, F., Hasekamp, O. P., Heymann, J., Kaminski, T., Laeng, A., Lichtenberg, G.,
De Mazière, M., Noël, S., Notholt, J., Orphal, J., Popp, C., Parker, R., Scholze, M., Suss-
mann, R., Stiller, G. P., Warneke, T., Zehner, C., Bril, A., Crisp, D., Griffith, D. W. T., Kuze,
A., O'Dell, C., Oshchepkov, S., Sherlock, V., Suto, H., Wennberg, P., Wunch, D., Yokota, T.,
and Yoshida, Y.: The Greenhouse Gas Climate Change Initiative (GHG-CCI): Comparison
and quality assessment of near-surface-sensitive satellite-derived $CO_2$ and $CH_4$ global data
sets, Remote Sensing of Environment, 162, 344–362, 2015.

Fraser, A., Chan Miller, C., Palmer, P. I., Deutscher, N. M., Jones, N. B., and Griffith, D. W. T.:
The Australian methane budget: Interpreting surface and train-borne measurements using a
chemistry transport model, Journal of Geophysical Research: Atmospheres, 116, n/a–n/a,
doi:10.1029/2011JD015964, http://dx.doi.org/10.1029/2011JD015964, d20306, 2011.

Jones, A., Walker, K. A., Jin, J. J., Taylor, J. R., Boone, C. D., Bernath, P. F., Brohede, S.,
Manney, G. L., McLeod, S., Hughes, R., and Daffer, W. H.: Technical Note: A trace gas
climatology derived from the Atmospheric Chemistry Experiment Fourier Transform Spec-
trometer (ACE-FTS) data set, Atmospheric Chemistry and Physics, 12, 5207–5220, doi:
10.5194/acp-12-5207-2012, http://www.atmos-chem-phys.net/12/5207/2012/, 2012.

Monge-Sanz, B. M., Chipperfield, M. P., Untch, A., Morcrette, J.-J., Rap, A., and Simmons,
A. J.: On the uses of a new linear scheme for stratospheric methane in global models:
water source, transport tracer and radiative forcing, Atmospheric Chemistry and Physics, 13,
9641–9660, doi:10.5194/acp-13-9641-2013, http://www.atmos-chem-phys.net/13/9641/2013/,
2013.

Park, R. J., Jacob, D. J., Field, B. D., Yantosca, R. M., and Chin, M.: Natural and transboundary
pollution influences on sulfate-nitrate-ammonium aerosols in the United States: Implications
for policy, Journal of Geophysical Research: Atmospheres, 109, 2004.

Patra, P., Krol, M., Montzka, S., Arnold, T., Atlas, E. L., Lintner, B., Stephens, B., Xiang, B.,
Elkins, J., Fraser, P., et al.: Observational evidence for interhemispheric hydroxyl-radical
parity, Nature, 513, 219–223, 2014.

Rodgers, C. D. and Connor, B. J.: Intercomparison of remote sounding instruments, Journal of
Geophysical Research, 108, 4116, doi:10.1029/2002JD002299, 2003.

Saad, K. M., Wunch, D., Toon, G. C., Bernath, P., Boone, C., Connor, B., Deutscher, N. M.,
Griffith, D. W. T., Kivi, R., Notholt, J., Roehl, C., Schneider, M., Sherlock, V., and Wennberg,
P. O.: Derivation of tropospheric methane from TCCON $CH_4$ and HF total column observations, Atmospheric Measurement Techniques, 7, 2907–2918, doi:10.5194/amt-7-2907-2014, http://www.atmos-meas-tech.net/7/2907/2014/, 2014.

Stephens, B. B., Gurney, K. R., Tans, P. P., Sweeney, C., Peters, W., Bruhwiler, L., Ciais, P., Ramonet, M., Bousquet, P., Nakazawa, T., Aoki, S., Machida, T., Inoue, G., Vinnichenko, N., Lloyd, J., Jordan, A., Heimann, M., Shibistova, O., Langenfelds, R. L., Steele, L. P., Francey, R. J., and Denning, A. S.: Weak Northern and Strong Tropical Land Carbon Uptake from Vertical Profiles of Atmospheric CO2, Science, 316, 1732–1735, doi:10.1126/science.1137004, http://science.sciencemag.org/content/316/5832/1732, 2007.

von Clarmann, T., Höpfner, M., Kellmann, S., Linden, A., Chauhan, S., Funke, B., Grabowski, U., Glatthor, N., Kiefer, M., Schieferdecker, T., Stiller, G. P., and Versick, S.: Retrieval of temperature, H$_2$O, O$_3$, HNO$_3$, CH$_4$, N$_2$O, ClONO$_2$ and ClO from MIPAS reduced resolution nominal mode limb emission measurements, Atmospheric Measurement Techniques, 2, 159–175, doi:10.5194/amt-2-159-2009, http://www.atmos-meas-tech.net/2/159/2009/, 2009.

Wunch, D., Toon, G. C., Wennberg, P. O., Wofsy, S. C., Stephens, B. B., Fischer, M. L., Uchino, O., Abshire, J. B., Bernath, P., Biraud, S. C., Blavier, J.-F. L., Boone, C., Bowman, K. P., Browell, E. V., Campos, T., Connor, B. J., Daube, B. C., Deutscher, N. M., Diao, M., Elkins, J. W., Gerbig, C., Gottlieb, E., Griffith, D. W. T., Hurst, D. F., Jiménez, R., Keppel-Aleks, G., Kort, E. a., Macatangay, R., Machida, T., Matsueda, H., Moore, F., Morino, I., Park, S., Robinson, J., Roehl, C. M., Sawa, Y., Sherlock, V., Sweeney, C., Tanaka, T., and Zondlo, M. a.: Calibration of the Total Carbon Column Observing Network using aircraft profile data, Atmospheric Measurement Techniques, 3, 1351–1362, doi:10.5194/amt-3-1351-2010, http://www.atmos-meas-tech.net/3/1351/2010/, 2010.

Wunch, D., Toon, G. C., Blavier, J.-F. L., Washenfelder, R. a., Notholt, J., Connor, B. J., Griffith, D. W. T., Sherlock, V., and Wennberg, P. O.: The Total Carbon Column Observing Network, Philosophical transactions. Series A, Mathematical, physical, and engineering sciences, 369, 2087–2112, doi:10.1098/rsta.2010.0240, http://www.ncbi.nlm.nih.gov/pubmed/21502178, 2011.

Wunch, D., Toon, G. C., Sherlock, V., Deutscher, N. M., Liu, C., Feist, D. G., and Wennberg, P. O.: The Total Carbon Column Observing Network's GGG2014 Data Version, Tech. rep., Carbon Dioxide Information Analysis Center, Oak Ridge National Laboratory, Oak Ridge, Tennessee, U.S.A. doi: 10.14291/tccon.ggg2014.documentation.R0/1221662, doi: 10.14291/tccon.ggg2014.documentation.R0/1221662, http://dx.doi.org/10.14291/tccon.ggg2014.
documentation.R0/1221662, 2015.
* * *
[Figure]

figure-1.pdf

**Fig. 1.** Detrended seasonality of TCCON (black diamonds), GEOS-Chem base (red circles), and GEOS-Chem aseasonal (blue squares) $CH_4$ column-averaged DMFs, averaged across Northern Hemisphere sites, except Bialystok, Bremen, Saga, and Réunion Island. Error bars denote the $1\sigma$ standard deviation across sites.

[Figure]

figure-2.pdf

**Fig. 2.** NOAA tall tower PFP flask (black dashed line), NOAA surface flask (black solid line), and GEOS-Chem surface level (red solid line) seasonality of $CH_4$ DMFs over 2005-2011 at Park Falls, WI, USA and Baring Head, NZ. PFP data is courtesy of Arlyn Andrews (NOAA): Andrews, A.E., E. Dlugokencky, and P.M. Lang (2008), Methane Dry Air Mole Fractions from the NOAA ESRL Surface Network using Programmable Flask Packages (PFP), 1992-2008, Version: 2013-07-03.

[Figure]

[Figure]

**Fig. 3.** Please see caption on Fig. 1.

[Figure]

[Figure]

**Fig. 4.** Please see caption on Fig. 2.

---

## Author Response (AR2)

**Author Response to Review of Referee #3 for "Seasonal Variability of Stratospheric Methane: Implications for Constraining Tropospheric Methane Budgets Using Total Column Observations"**

K. M. Saad[1], D. Wunch[1,2], N. M. Deutscher[3,4], D. W. T. Griffith[3], F. Hase[5], M. De Mazière[6], J. Notholt[4], D. F. Pollard[7], C. M. Roehl[1], M. Schneider[5], R. Sussmann[8], T. Warneke[4], and P. O. Wennberg[1]

[1]California Institute of Technology, Pasadena, California, USA
[2]University of Toronto, Toronto, Ontario, Canada
[3]University of Wollongong, Wollongong, NSW, Australia
[4]University of Bremen, Bremen, Germany
[5]Karlsruhe Institute of Technology, IMK-ASF, Karlsruhe, Germany
[6]Royal Belgian Institute for Space Aeronomy, Brussels, Belgium
[7]National Institute of Water and Atmospheric Research, Omakau, New Zealand
[8]Karlsruhe Institute of Technology, IMK-IFU, Garmisch-Partenkirchen, Germany

*Correspondence to:* K. M. Saad (katsaad@caltech.edu)

We thank Referee #3 for their additional comments.

In addition to the comments replied to below, two of the papers referenced have been changed: Ostler et al. (2015) has been updated to the published version, Ostler et al. (2016), and Toon and Wunch (2014) has been substituted for Wunch et al. (2015) as the more appropriate reference for the TCCON priors in Section 2.2.1.

**P 2, L29-33: I would rephrase the characterization of the Stephens et al. (2007) findings. Saying that the NH sink and tropical source were found to be overestimated "when constraining models with aircraft CO2 profiles" really makes it sound as if the aircraft profiles were assimilated, which isn't the case.**

The wording has been changed to, " [...] when comparing models to aircraft $CO_2$ profiles" to avoid mischaracterizing

10    Stephens et al. (2007).

**P3, L13: These forward model dependencies of CH4 concentrations \*on\* vertical transport**

The preposition in this sentence has been updated.

15   **P3, L23: emissions -> emission OR emissions fluxes -> emissions OR emissions fluxes -> fluxes**

This sentence has been changed to, " [...] we compare the base case simulation to one in which emissions do not vary within each year [...]"

**P6, L7-11: Both reviewers commented on the oddness of this approach for deriving pseudo-aseasonal fluxes, which you**

**explained as being due to a technical limitation of the model setup. Please include this justification here as well, and not only in the Conclusions. Perhaps you could refer to Figure 13 in the appendix as well.**

The following sentences have been added to p.6 l.11: "The model infrastructure posed difficulties for setting the seasonally-varying fluxes constant throughout each year; thus we implement this scaling technique as an alternative to assess first-order impacts of emission seasonality. The resulting changes to the spatial distribution of $CH_4$ emissions are shown in Fig. 13."

**P6, L12-15: Reads a bit awkwardly and as a result is a bit unclear. Perhaps break it into two sentences, or rewrite in some way.**

This sentence has been divided into two sentences to improve clarity:

"For comparisons with column measurements, model vertical profiles were smoothed with corresponding TCCON $CH_4$ averaging kernels, interpolated for the daily mean solar zenith angles, and prior profiles, scaled with daily median scaling factors, following the methodology in Rodgers and Connor (2003) and Wunch et al. (2010). Averaging kernels and prior profiles were interpolated to the model's pressure grid, and all terms in the smoothing equation were interpolated to daily mean surface pressures measured at each site."

**P8, L5: in the both the base and -> in both the base and**

The extraneous "the" has been removed.

**P12, Figure 7 caption: Reunion is not in the Northern Hemisphere, which is implied here.**

The inclusion of Réunion Island in the figure caption was an oversight, as Saga and Réunion Island were the only sites with less than a year of measurements a seasonal cycle. The caption now reads, "[...] averaged across Northern Hemisphere sites, excluding Saga, which has less than one year of measurements prior to 2012."

**P14, L11-12: "The model's stratospheric response to emissions perturbations differ from that of the troposphere and are subject to different transport and loss errors." -> "The model's stratospheric response to emission perturbations differs from that of the troposphere and is subject to different transport and loss errors."**

The grammatical numbering in this sentence has been changed from plural to singular.

**P15: in text on Figure 9, change "Emissions Seasonality" to either "Emissions' Seasonality" or "Emission Seasonality"**

The legend on Figure 9 now reads "Tropospheric Dependence on Emission Seasonality."

**P15, L2: emissions errors -> emission errors OR emissions' errors**

The wording has been changed to "emission errors" on this line, as well as on p. 10 l.6.

**P15, L5-6: "These posterior emissions are scaled for each sector according to their a priori fraction of total emissions in each grid box." Well, it depends a bit on the inversion set-up. Not every system does it quite like this, it depends also on the a prior uncertainties assigned to each sector/process. I don't disagree with the general conclusions, but if you're going to describe a specific inversion approach, please reference it as such. It is possible to not specify the seasonal cycle in the prior, but have it driven by the data alone.**

This phrasing was intended to provide a simplified description of how underlying assumptions about the spatial distribution of a priori emissions for a given sector relative to those of other sectors will impact posterior emissions. In optimizing total $CH_4$ emissions, we agree that the spatial distribution and seasonality is driven primarily by assimilated data (assuming correct transport and OH fields). However, even if seasonal cycles for various emissions sectors are not specified a priori, the inversion must include some assumptions about how different sectors are scaled relative to each other for a given data-driven adjustment. In addition, the uncertainties for $CH_4$ emissions are sufficiently large and unknown that many inversions assign a fixed percentage error for each sector that does not add information about the sectors' relative contributions.

We acknowledge that the phrasing of this sentence could be misunderstood for a delineation of inversion methodology, without consideration of more sophisticated inversion techniques, and have changed it to, "Attribution of these posterior emissions to different sectors depends on a priori information and assumptions about how they vary in time and location relative to one another."

**P 15: "derived by calculating the total emissions resulting from an increase of 1 ppb of CH4 in each tropospheric column and scaling". I better understood what you did after reading the response to my comment. Please integrate this information into the manuscript so that other readers may also benefit from this clarification (explaining, for instance, that transport is not considered, which is a critical limitation of the approach). And please, reword this sentence to say "the total emissions required to produce an increase of 1 ppb..."**

The additional clarification has been inserted into the text, and the suggestion for the parenthetical on l.9 has also been incorporated:

> "For example, Fig. 10b illustrates the sensitivity of posterior wetland emissions to a three-month lag in the Northern Hemisphere. The change in posterior emissions is derived by calculating the total emissions required to produce an increase of 1 ppb of $CH_4$ in each tropospheric column and scaling those emissions according to the a priori contribution of wetlands, estimated as the fractional contribution of wetlands to the total monthly mean emissions. The difference between this change in wetland emissions and the value in the same location three months prior produces the sensitivity of wetland emissions to the tropospheric phase lag. This approach provides an alternative to the computationally expensive calculation of the gain matrix over the entire time series but does not include information about model transport."

**P17, L35: scaled -> scales**

This wording has been updated.

5 **P21, L6: wrong style of reference to Wunch et al. (2015)**

The citation has been corrected to be in-text instead of parenthetical.

**P23, L3: as reference -> for reference**

This wording has been updated.

[revised manuscript text omitted]
. Attribution of these posterior emissions to different sectors depends on a priori information and assumptions about how they vary in time and location relative to one another. Thus, an increase in posterior emissions relative to the prior in the northern mid and high latitudes during winter will not change emissions from wetlands. For example, Fig. 10b illustrates the sensitivity of posterior wetland emissions to a three-month lag in the Northern Hemisphere. The change in posterior emissions is derived by calculating the total emissions  required to produce an increase of 1 ppb of $CH_4$ in each tropospheric column and scaling those emissions according to the a priori contribution of wetlands estimated as the fractional contribution of wetlands to the total monthly mean emissions. The difference between this change in wetland emissions and the value in the same location three months prior produces the sensitivity of wetland emissions to the

[Figure]

**Figure 10.** (a) GEOS-Chem monthly zonal mean wetland emissions, in Gg. (b) The Northern Hemisphere sensitivity of GEOS-Chem wetland emission attribution caused by a 3-month lag for each 1 ppb increase of $CH_4$ in the tropospheric column, in Gg.

tropospheric phase lag. This approach provides an alternative to the computationally expensive calculation of the gain matrix over the entire time series but does not include information about model transport.

[revised manuscript text omitted]
_{CH_4}^s = \gamma_{CH_4} \cdot X_{CH_4}^a + \boldsymbol{a}_{\mathbf{CH_4}}^{\S}(\boldsymbol{x}_{\mathbf{CH_4}}^{\boldsymbol{m}} - \gamma_{CH_4}\boldsymbol{x}_{\mathbf{CH_4}}^{\boldsymbol{a}}) \tag{B2}$$

where $X_{CH_4}^s$ is the smoothed GEOS-Chem column-averaged DMF, $\gamma_{CH_4}$ is the TCCON daily median retrieved profile scal-
10   ing factor, and $\boldsymbol{x}_{\mathbf{CH_4}}^{\boldsymbol{a}}$ and $X_{CH_4}^a$ are respectively the a priori profile and column-integrated $CH_4$ DMFs (Rodgers and Connor, 2003). The pressure weighting function, $\boldsymbol{h}$, was applied such that $X = \boldsymbol{h}^T\boldsymbol{x}$. TCCON priors were interpolated to the GEOS-Chem pressure grid, and GEOS-Chem pressure and corresponding gas profiles were adjusted using daily mean surface pressures local to each site (Wunch et al., 2010; Messerschmidt et al., 2011). The averaging kernels were interpolated for the local daily mean solar zenith angle and the GEOS-Chem pressure grid so that it could be applied to the difference between the
15   GEOS-Chem and TCCON profiles as $\boldsymbol{a}^{\S}\boldsymbol{x} = \sum_{i=1}^{N} a_i h_i x_i$ from the surface to the highest level, $N$, at $i$ pressure levels (Connor et al., 2008; Wunch et al., 2011b). Figure 16 shows how the smoothed column compares to the column that only uses the dry gas correction.

[Figure]

**Figure 16.** GEOS-Chem smoothed versus dry integrated $CH_4$ DMFs for base simulation tropospheric columns, total columns, and stratospheric contribution. Site colors are as in Fig. 1. Dashed lines mark the one-to-one lines.

**Appendix C:  Derivation of Stratospheric Contribution**

Considering the $CH_4$ profile integration as in Equation A4, and substituting the profile of $CH_4$ in the stratosphere, $x_{CH_4}(P) = x_{CH_4}(P^t) + \delta \cdot P$, described in Appendix A, the total column is calculated as:

$$\int_0^{P^s} x_{CH_4} \frac{dP}{gm} = \int_{P^t}^{P^s} x_{CH_4} \frac{dP}{gm} + \int_0^{P^t} [x_{CH_4}(P^t) + \delta \cdot P] \frac{dP}{gm} \tag{C1}$$

$$X_{CH_4} \cdot P^s = X_{CH_4}^t [P^s - P^t] + x_{CH_4}(P^t) \cdot P^t + c_{CH_4}^\delta \tag{C2}$$

where $c_{CH_4}^\delta$, is the pressure-weighted column average of $CH_4$ loss in the stratosphere. Rearranging terms, Equation C2 becomes:

$$[X_{CH_4} - X_{CH_4}^t] P^s = [x_{CH_4}(P^t) - X_{CH_4}^t] P^t + c_{CH_4}^\delta \tag{C3}$$

[revised manuscript text omitted]